# Muscle-derived Myoglianin regulates *Drosophila* imaginal disc growth

Ambuj Upadhyay, Aidan J Peterson, Myung-Jun Kim, Michael B O'Connor*

Department of Genetics, Cell Biology and Development University of Minnesota, Minneapolis, United States

**Abstract** Organ growth and size are finely tuned by intrinsic and extrinsic signaling molecules. In *Drosophila,* the BMP family member Dpp is produced in a limited set of imaginal disc cells and functions as a classic morphogen to regulate pattern and growth by diffusing throughout imaginal discs. However, the role of TGFβ/Activin-like ligands in disc growth control remains ill-defined. Here, we demonstrate that Myoglianin (Myo), an Activin family member, and a close homolog of mammalian Myostatin (Mstn), is a muscle-derived extrinsic factor that uses canonical dSmad2-mediated signaling to regulate wing size. We propose that Myo is a myokine that helps mediate an allometric relationship between muscles and their associated appendages.

## Introduction

How organ growth is regulated to achieve proper scaling with final body size has been a long-standing question in developmental biology. Organ growth is a dynamic process that is finely tuned by highly conserved signaling molecules that are produced and act within the tissue of origin (intrinsic) or on a distant tissue (extrinsic) (*Boulan et al., 2015*; *Hariharan, 2015*; *Johnston and Gallant, 2002*; *Mirth and Shingleton, 2012*). These growth signals regulate final organ size through three distinct processes: cell division (proliferation), mass accumulation (cell growth) and cell death (apoptosis). The *Drosophila* wing imaginal disc, the larval precursor to adult wings and thorax, has been widely used as a model to study the molecular mechanisms that control organ growth and size control (*Martín et al., 2009*). Genetic analysis has identified numerous wing disc intrinsic factors, such as Wnt, Hedgehog, and Bone Morphogenetic Protein (BMP) homologs, that are required for proper patterning and proliferation (*Hariharan, 2015*). In contrast, examples of extrinsic factors that regulate imaginal disc growth are limited. The insulin-like signaling pathway is one clear exception. Insulin-like peptides are produced in the brain and fat body and are well known to promote both growth and proliferation in many tissues including imaginal disc cells (*Boulan et al., 2015*; *Droujinine and Perrimon, 2013*; *Grewal, 2012*; *Nijhout, 2003*; *Shingleton et al., 2007*; *Sopko and Perrimon, 2013*; *Teleman, 2010*). Insulin-like factors act in a physiological capacity to coordinate overall organismal growth with nutrition. Identification of other extrinsic growth signals could provide new insights into mechanisms governing coordinated organ scaling.

The largest and most diverse group of signaling molecules is the Transforming Growth Factor β (TGFβ) superfamily of growth and differentiation factors. The superfamily is divided into the BMP and the Activin/TGFβ-like subfamilies based on sequence and signaling relationships. In higher vertebrates, these growth factors play a myriad of roles during development from cell fate specification and stem cell maintenance to control of cellular growth and proliferation (*Morikawa et al., 2016*). Both TGFβ branches are conserved in *Drosophila*, where limited functional redundancy and a powerful genetic toolkit provide an attractive system for study (*Peterson and O'Connor, 2014*; *Upadhyay et al., 2017*). The *Drosophila* BMP ligand Decapentaplegic (Dpp) functions as a classic intrinsic morphogen, since it is secreted from a limited number of imaginal disc source cells and then diffuses throughout the tissue to regulate proliferation and patterning based on its concentration

*For correspondence:
moconnor@umn.edu

Competing interests: The authors declare that no competing interests exist.

gradient (*Campbell and Tomlinson, 1999*; *Lecuit et al., 1996*; *Restrepo et al., 2014*). Conversely, the Activin-like branch also plays a significant role in disc growth, however, it is much less characterized (*Brummel et al., 1999*; *Hevia and de Celis, 2013*; *Hevia et al., 2017*).

In *Drosophila*, Activin-like signaling is initiated by the three ligands, Myoglianin (Myo), Activinβ (Actβ), and Dawdle (Daw). These ligands bind to and activate a hetero-tetrameric receptor complex on the cell membrane consisting of two type-I receptor subunits encoded by the *baboon (babo)* locus and two type-II receptor subunits encoded by either the *punt* or *wishful-thinking (wit)* genes or perhaps a combination of the two (*Upadhyay et al., 2017*). Activated Babo phosphorylates cytosolic dSmad2 which then accumulates in the nucleus and, along with the co-Smad Medea, regulates expression of target genes (*Brummel et al., 1999*; *Hevia et al., 2017*).

Notably, there are three splice isoforms of *babo* with different ligand binding domains, which presumably provide ligand binding specificity (*Jensen et al., 2009*). *Babo* null mutants die at the larval-pupal transition and have significantly smaller wing imaginal discs and escapers, produced by hypomorphic alleles, also have smaller adult wings (*Brummel et al., 1999*). Conversely, expression of constitutively activated Babo in wing imaginal discs leads to larger adult wings that contain more cells, suggesting that Activin-like signaling has a role in regulating proliferation (*Brummel et al., 1999*; *Hevia and de Celis, 2013*). Additionally, Babo can signal non-canonically through Mad (the BMP Smad) to stimulate wing disc overgrowth, but only in the background of a protein null allele of *dSmad2* (*Peterson and O'Connor, 2013*).

For both the dSmad2 canonical and the Mad non-canonical signaling pathways, it remains unclear which Babo isoform and which Activin-like ligands, or combination thereof, regulates disc growth, nor is it clear which tissue(s) provide the ligands that promote disc growth. Expression analysis by in situ hybridization or RNAseq data suggests that all Activin-like ligands are expressed in the imaginal wing disc, albeit at low levels, perhaps indicating collective contribution to stimulating tissue growth (*Graveley et al., 2011*; *Hevia and de Celis, 2013*). However it is important to recognize that mRNA expression does not necessarily correlate with secretion of functional ligand, especially since TGFβ−type ligands undergo a series of post-translational modifications and ligand processing to generate active molecules and these processes may vary within different tissues (*Akiyama et al., 2012*).

Here, we report our analysis of Activin-like ligands and Baboon isoforms for their roles in regulating imaginal disc growth. Through a series of genetic experiments, we demonstrate that among the three *Drosophila* Activin-like ligands, only Myo is required for proper growth of imaginal discs. We also identify the receptors Babo-A and Punt as being necessary for Myo regulation of wing imaginal disc growth via phosphorylation of dSmad2. Surprisingly, we identify the tissue source of Myo that controls disc growth to be the larval muscle.

## Results

### *Myoglianin* is required for proper disc growth

All *Drosophila* Activin-like ligands (Myo, Actβ, and Daw) have tissue specific function yet signal via the sole Type-I receptor Babo. Hence, the *babo* mutant phenotype is likely a composite of the three individual ligand mutant phenotypes. The imaginal discs of *babo* mutants are approximately 30% smaller than wild-type but do not exhibit any obvious defects in tissue patterning (*Brummel et al., 1999*). We set out to identify which Activin-like ligand(s) are required for proper imaginal disc growth. Null *myo*, *Actβ*, and *daw* mutants are larval or pharate lethal (*Awasaki et al., 2011*; *Ghosh and O'Connor, 2014*; *Zhu et al., 2008*; *Moss-Taylor et al., 2019*). Therefore, we analyzed the size of imaginal wing discs from late third instar larvae of null mutant animals. For this initial analysis, we used a simple yeast/sugar diet (5Y5S), because *daw* mutants are sensitive to standard acidic cornmeal food (*Ghosh and O'Connor, 2014*). Interestingly, only *myo* mutants yield significantly smaller discs, phenocopying *babo* mutants. *Myo* wing discs are approximately 40% smaller than $w^{1118}$ controls, but retain their normal shape and pattern (*Figure 1B and E*). *Actβ* mutant discs had no significant size defects, while *daw* mutant discs were slightly smaller, which may be attributable to the reduction of insulin secretion in the *daw* mutants (*Figure 1C-E*; *Ghosh and O'Connor, 2014*). Previous studies suggested that all Activin-like ligands are expressed in the wing disc (*Hevia and de Celis, 2013*). Despite this observation, our result clearly suggests that only *myo* is functionally

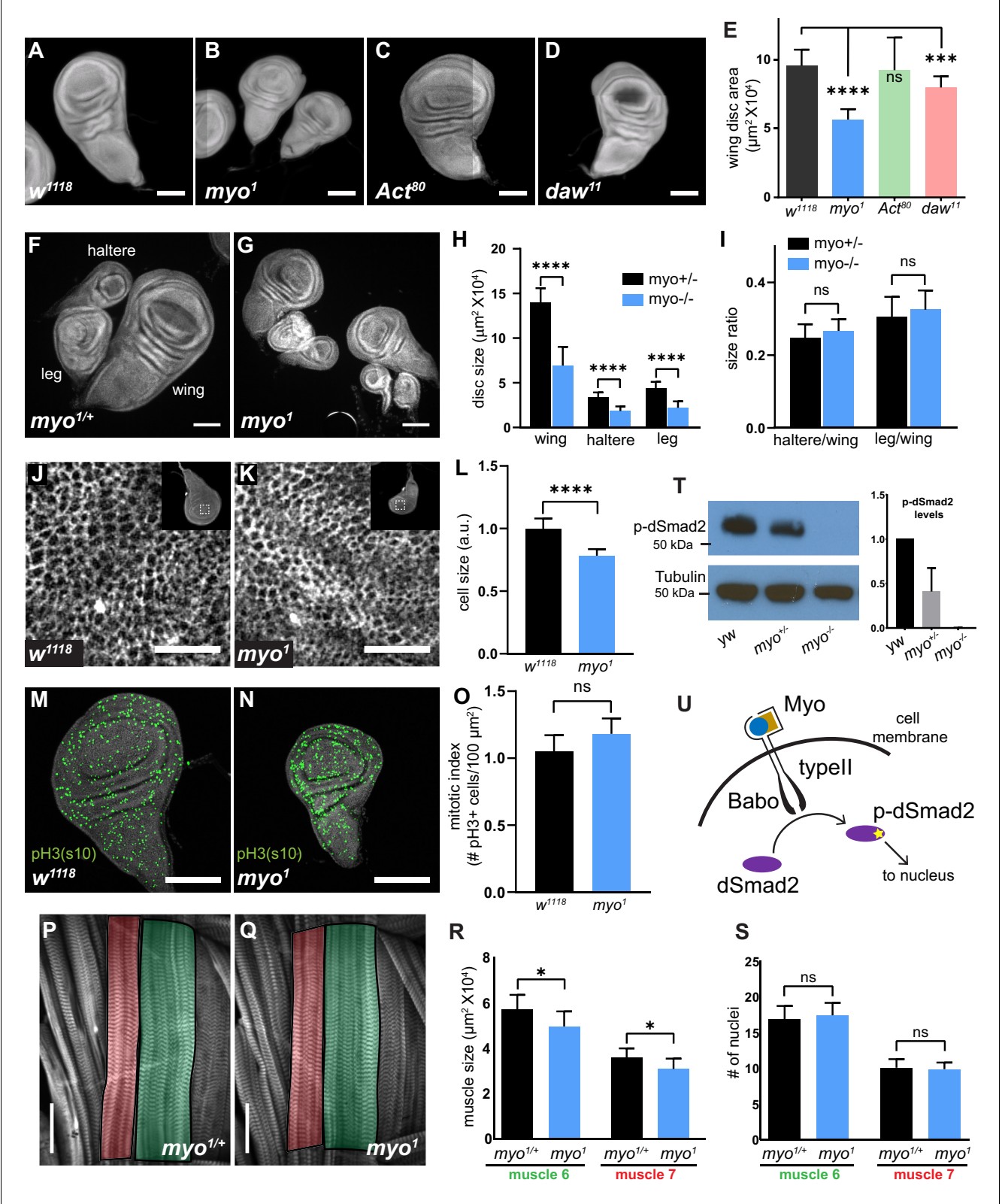

**Figure 1.** *Myoglianin* regulates size of imaginal discs and activates dSmad2. (A–D) DAPI staining of wing imaginal discs from late third instar larvae of various TGFβ ligand mutants. Scale bar, 100 μm. (E) Wing disc size of *myo* mutants (blue bar) are approximately 40% smaller than control, *Activinβ* mutant discs (C) are the same size as controls (green bar), and *dawdle* mutant discs (D) are ~16% smaller (red bar) (n > 20 per group). (F–I) Wing, leg, and haltere imaginal discs of *myo¹* null mutants are 50% smaller vs heterozygous *myo¹/⁺* controls. (H) Size of various imaginal discs, normalized to

*Figure 1 continued on next page*

*Figure 1 continued*

controls. (I) Size ratio of haltere/wing disc (.25) or leg/wing disc (.3) are the same in *myo* mutants vs controls. (J–K) Phalloidin staining showing the apical actin-belt of epithelial cells in wing discs; each cell has a distinct actin belt that defines the edges of a cell (scale bar = 10 μm, n > 5 per group). Single confocal section is shown with the whole disc in insert with the magnification of the pouch region outlined. (L) Based on cell density, *myo* mutant discs cells are 12% smaller vs control. (M–N) Phospho-Histone H3-pSer10 (pH3s10) staining to mark mitotic cells in early third instar wing discs, showing no difference in number of mitotic cells in *myo* mutants (scale bar = 100 μm, n = 5 per group). Number of mitotic cells per 100 $μm^2$ is plotted (O). (P–Q) Phalloidin staining showing the late 3$^{rd}$ instar larvae body wall muscle #6 (green highlight) and #7 (red highlight) (scale bar = 100 μm, n = 10). Muscle area (R) and number of nuclei per fiber (S) is plotted. *myo* mutant muscles are 13% smaller (p=0.02) yet have the same number of nuclei per fiber (S). (T) Western blot analysis of phospho-dSmad2 levels in wing discs, with quantification from 3 separate blots. *myo* mutants lack p-dSmad2 signal indicating loss of canonical signaling. (U) Model for canonical TGFβ signaling in wing imaginal disc via Myo/Babo/dSmad2. C-terminal activation of dSmad2 (yellow star) results in regulation in expression of downstream target genes which is required for proper growth of imaginal discs.

The online version of this article includes the following figure supplement(s) for figure 1:

**Figure supplement 1.** *myo* mutants show no defects in cell death or cell growth.
**Figure supplement 2.** Trans-heterozygote *myo* mutants do not have larger muscles.

required for growth of the discs, and that other ligands may have other functions apart from disc growth. We next asked whether other imaginal tissues such as the haltere and leg discs are also affected by the loss of *myo*. Indeed, both these imaginal discs are also approximately ~40% smaller in *myo* mutants and maintain their proportional size (*Figure 1F-I*). This suggests that Myo signaling regulates tissue growth in all three of these imaginal discs to similar extents.

To determine why *myo* mutant discs are small, we measured cell size, cell growth, proliferation, and apoptosis. First, to assess cell size, we used phalloidin staining to mark the apical actin belt, and measured the cell density at the epithelium apical surface. We observed that the cells in *myo* mutants are more densely packed. We calculated that the cells in these mutants are approximately ~20% smaller than controls (*Figure 1J-L*). This observation is consistent with the smaller cell size report for *dSmad2* mutant discs (*Peterson and O'Connor, 2013*). Next, we checked for cell proliferation and cell death markers at 90–96 hr after egg lay (AEL ), a time at which a large amount disc growth occurs, and because cell death pathways are blocked later in development (*Jaszczak and Halme, 2016*). We examined proliferation by staining for phospho-Histone 3(ser10) to mark mitotic cells, and found no significant difference between *myo* discs and controls at this stage (*Figure 1M-O*). Next, we stained 90–96 hr AEL discs with anti-cleaved-caspase 3, an apoptosis marker, and found no increase in apoptosis in *myo* mutants (*Figure 1—figure supplement 1, A–B*).

Another possible explanation for why *myo* mutant discs do not grow properly could be defects in TOR pathway regulation. To address this issue, we stained growing early L3 imaginal discs for phospho-S6, which is a target of S6K and a readout of TOR pathway activation (*Romero-Pozuelo et al., 2017*). We found no decrease in pS6 staining in *myo* mutant discs, indicating that TOR signaling is not perturbed in these cells (*Figure 1—figure supplement 1, C–D*). We conclude that *myo* mutant discs are smaller in part due to smaller cells. However, the 22% decrease in cell size is insufficient to completely account for the ~40% decrease in total tissue size. Because our proliferation and apoptosis assays are snapshot experiments of fixed tissues, we infer that the portion of final disc size that cannot be accounted for by smaller cell size represents either a small proliferation reduction or a slight increase in apoptosis rate that cannot be detected at individual time points but, over the course of the entire developmental period, leads to an additional ~20% decrease in disc size. We note that a previous study of *dSmad2* mutant clones in wing discs reported smaller *dSmad2* mutant average clone size compared to control clones, consistent with a reduction in proliferation rate (*Hevia et al., 2017*).

*Myo* is homologous to vertebrate *Mstn*, which negatively regulates skeletal muscle size by inhibiting proliferation of muscle stem cells (*Morikawa et al., 2016*) and regulation of protein homeostasis (*Lee et al., 2011*; *Rodriguez et al., 2014*; *Trendelenburg et al., 2009*). Recent studies in *Drosophila* using RNAi tissue-specific knockdown suggested that Myo also functions as a negative regulator of muscle size similarly to vertebrate *Mstn* (*Augustin et al., 2017*). We re-examined this issue using *myo* null alleles and measured the size of muscles 6 and 7, as well as the number of nuclei per fiber from the 2nd Abdominal segment. Surprisingly, *myo$^1$* mutant muscles are 13% smaller vs heterozygote controls (p=0.02), however they contain the same number of nuclei per fiber, which suggests that there were no defects in myoblast fusion during embryogenesis (*Figure 1—figure supplement*

*1, P–S*). We repeated the experiment using a newly generated *myo^CR2* allele and analyzed *myo^{1/CR2}* transheterozygotes to rule out 4^th chromosome background effects. In this case, we find no difference in muscle size compared to controls again indicating that *myo* mutants do not have larger muscles compared to heterozygote controls (*Figure 1—figure supplement 2*). We also repeated the RNAi experiments and find no statistical difference in muscle size of *mef2>myo* miRNA vs *w1118; myo-miRNA* controls (*Figure 1—figure supplement 2*). Additionally, ubiquitous knockdown of *myo* using *da-GAL4* does not result in larger muscles compared to the *myo* null mutants, or *myo* heterozygotes (*Figure 1—figure supplement 2*). Taken together our results suggest that *Drosophila myo* is not functionally conserved with vertebrate *Mstn* in regulating muscle size.

## Myo signals canonically via dSmad2

Canonical TGFβ/Activin pathway activation in *Drosophila* is mediated by the Type-I receptor Baboon phosphorylating cytosolic dSmad2 at the C-terminus (*Brummel et al., 1999*; *Peterson and O'Connor, 2014*; *Upadhyay et al., 2017*). To determine if Myo acts through the canonical signaling pathway, we examined phosphorylated dSmad2 (p-dSmad2) levels by western blots of late L3 larvae wing disc extracts of *yw*, *myo* heterozygotes and null animals. In homozygous null mutants there is no signal under our detection conditions. These findings indicate that Myo is the primary ligand for TGFβ/Activin signaling in wing discs (*Figure 1T-U*).

## Myo signaling in imaginal discs requires a specific isoform of Baboon

*Baboon* is the sole Activin branch Type I receptor in *Drosophila*, but encodes three splice isoforms (*Awasaki et al., 2011*; *Jensen et al., 2009*). The three splice isoforms are a result of mutually exclusive alternative splicing of the fourth exon (4a, 4b, 4 c) giving rise to three receptors (Babo-A, Babo-B, and Babo-C) (*Brummel et al., 1999*; *Jensen et al., 2009*) that differ only in their extracellular, ligand binding domains (*Figure 2A*). It has been suggested that different tissues express different isoforms, perhaps in order to tune tissue-specific responses to various ligands (*Awasaki et al., 2011*; *Jensen et al., 2009*). Ectopic Myo expression in the larval brain is lethal, however viability can be partially restored by simultaneously knocking down *babo-a* but not *babo-b* or *babo-c*, which suggests that Myo signals via Babo-A (*Awasaki et al., 2011*). Since only *myo* is required for proper growth of imaginal discs (*Figure 1B*), we hypothesized that *babo-a* should be the major splice isoform expressed in the discs. We isolated late L3 wing discs and performed qPCR analysis using isoform specific primers, and detected only *babo-a* expression in the discs (*Figure 2B*). We verified that the primers for *babo-a*, *babo-b* and *babo-c* all amplify with similar efficiencies using known levels of control cDNA templates, and also found that each isoform shows different tissue-specific expression profiles (*Figure 2—figure supplement 1*). For example, the brain primarily expresses *babo-a* similar to the wing disc, while the fat body and gut primarily express *babo-c* and the carcass (muscle and epidermis) shows mixed lower expression levels of all three isoforms.

Next, we functionally tested whether isoform Babo-A is required for wing growth using the GAL4/UAS system to conduct tissue and isoform-specific RNAi knockdown experiments. Using tissue specific and distinct GAL4 drivers such as: *esg* (whole disc), *ci* (anterior compartment), and *hh* (posterior compartment) GAL4 lines, we knocked down individual *babo* isoforms and measured overall tissue size and cell size/density. In support of the qPCR data, we find that knocking down *babo-a* alone with *esg-GAL4* results in 40% smaller adult wings, whereas knocking down *babo-b* or *babo-c* has no effect on wing size (*Figure 2C-F, O*). We confirmed that the *babo-b* and *babo-c* RNAi constructs are active since, when expressed ubiquitously, they replicate *Actβ* and *daw* mutant phenotypes respectively. Using compartment specific GAL4 lines, we find that *Ci-GAL4* driving *babo-a* RNAi reduces the anterior compartment by 50% (*Figure 2F-I, L*) G-J,O). Next, knockdown of *babo-a* using the *hh-GAL4* is pupal lethal, thus we analyzed the late L3 imaginal wing discs. To measure the size of the posterior compartment in wing discs we co-expressed GFP. Similar to the other drivers, we observed that knocking down *babo-a* in the posterior compartment reduced the relative size of the compartment by 35% (*Figure 2K-N, P*). Surprisingly, we also note that, for unknown reasons, the normally straight anterior/posterior boundary (*Figure 2K*) becomes wavy when *babo-a* is knocked down with *hh-GAL4* (*Figure 2L*).

Since genetic loss of *myo* affected wing disc cell size, we also assessed adult cell size upon RNAi knockdown of *babo-a* by measuring trichome density in the adult wings. Each trichome on the wing

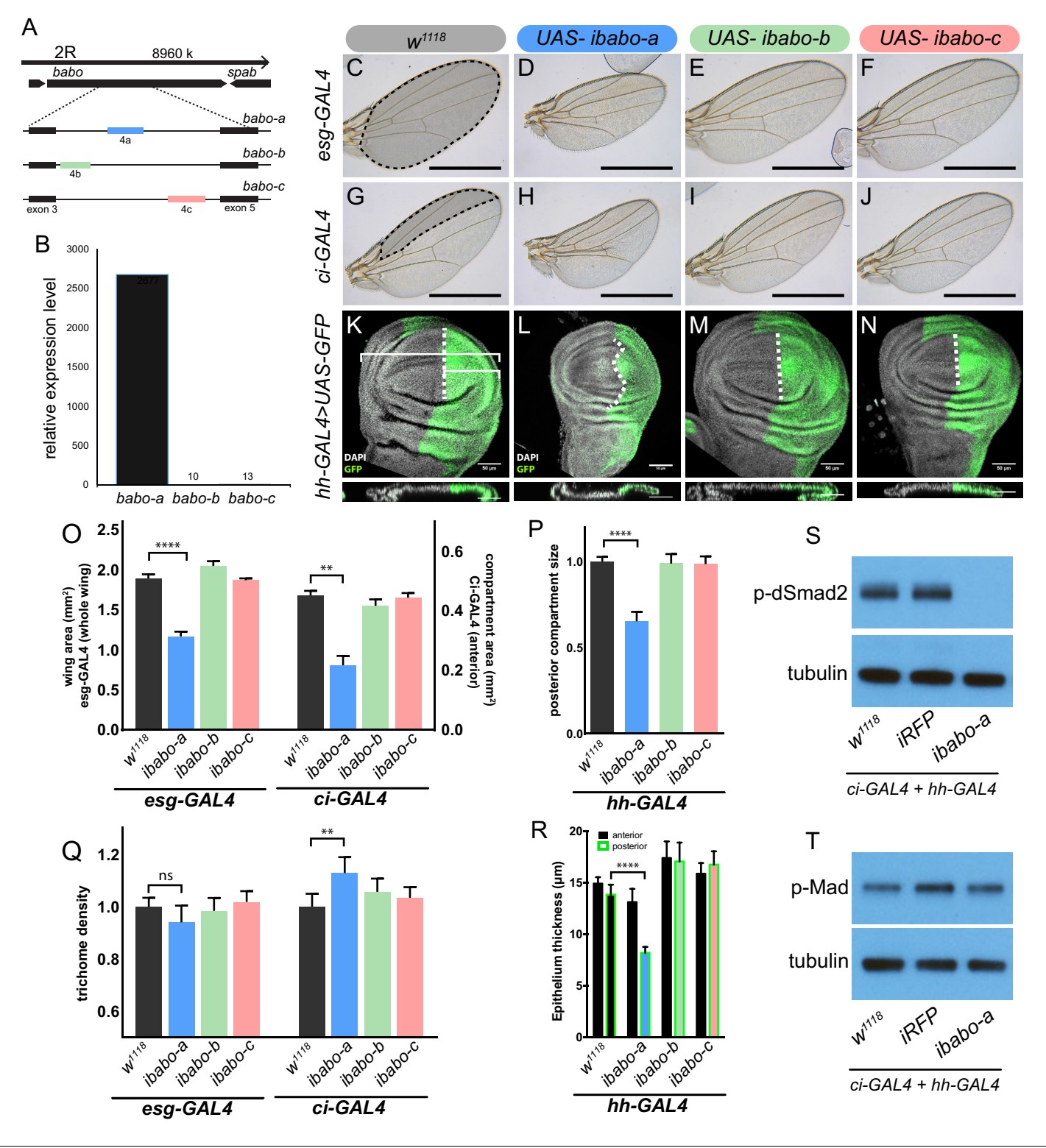

**Figure 2.** Babo-A regulates wing disc growth. (**A**) Cartoon illustrating the gene region and splice isoforms of TGFβ/Activin type-I receptor *babo*. The three splice isoforms are the result of 3 different mutually exclusive 4th exons; the remaining exons (not shown) are the same for all *babo* isoforms. (**B**) qPCR analysis of *babo* isoform expression from late 3rd instar wing imaginal discs identifies only *babo-a* expression. Expression level is relative to *Rpl32*, the expression level of *babo-b* and *babo-c* are indicated by the numbers above their respective bars. (**C–J**) Adult wings of animals where specific *babo* isoforms were knocked down in the entire wing blade (*esg-GAL4*) (**C–F**) or one compartment (*ci-GAL4*) (**G–J**). Knockdown of *babo-a* by *esg-GAL4*

*Figure 2 continued on next page*

Figure 2 continued

results in a smaller wing blade (D) and by *ci-GAL4* results in a smaller anterior compartment (H), scale bar 1 mm, n = 5–12 per group. Size measurements of tissue size (indicated by shading in C and G) are quantified in O. (K–N) Imaginal discs from late 3rd instar larvae where specific *babo* isoforms were knocked down in only the posterior compartment by *hh-GAL4*, marked with GFP, the size of the posterior compartment is measured relative to the entire wing pouch as illustrated by the brackets, the anterior/posterior boundary is highlighted by the dashed line, scale bar = 50 μm, n = 3–6 per group. (O) Quantification of tissue size from C-J. Knockdown of *babo-a* by *esg-GAL4* and *ci-GAL4* results in 40% and 50%, respectively, smaller tissues (O, blue bars). (P) Quantification of the relative size of the posterior compartment, data normalized to controls. Knockdown of *babo-a* by *hh-GAL4* results in a 35% reduction of the posterior compartment relative to the overall size of the wing disc. (Q) Quantification of trichome density in the affected region of adult wings from C-J, measurements are normalized to their respective GAL4 x *w1118* controls. Knockdown of *babo-a* using *esg-GAL4* does not affect trichome density, whereas knockdown with *ci-GAL4* results in higher trichome density, indicating smaller cells (P, blue bar). (R) Epithelium thickness was quantified from orthogonal views of the imaginal discs from K-N. Knockdown of *babo-a* results in a 29% decrease in epithelium thickness of the posterior compartment. (S–T) Western blot analysis of phospho-dSmad2 and phospho-Mad in L3 imaginal discs. Knockdown of *babo-a* throughout the imaginal discs using both *ci-GAL4* and *hh-GAL4* leads to loss of phospho-dSmad2 signal (S), but not phospho-Mad signal (T). Tubulin is shown as a loading control. n = 10 per lane.

The online version of this article includes the following figure supplement(s) for figure 2:

**Figure supplement 1.** qPCR analysis of Babo isoform expression in various tissues from late third instar larvae.

blade arises from a single wing disc epithelial cell. Therefore, by measuring the trichome density, relative cell size can be calculated. Knockdown of *babo-a* using *esg-GAL4* does not affect trichome density, whereas *ci-GAL4* result in a 13% increase in trichome density, i.e. smaller cells (*Figure 2Q*).

Another way that cell size could be affected is by altering the thickness of the disc epithelium. If the disc epithelium is thin then the individual cells are likely smaller. Using confocal optical sectioning we observed that knocking down *babo-a* leads to an ~40% decrease in the epithelium thickness (*Figure 2K-N*, (*cross-section*), *R*).

Previous studies have shown that Babo activates dSmad2 in numerous different tissues. Additionally, vertebrate TGFβ/Activin type-I receptors have also been shown to cross talk with BMP R-Smads. We hypothesized that under wild type conditions, Myo/Babo signaling regulates disc growth via phosphorylation of dSmad2 but not Mad. As predicted, knocking down *babo-a* in the entire wing disc by simultaneously using *ci-GAL4* and *hh-GAL4* completely abolishes phospho-dSmad2 levels (*Figure 2—figure supplement 1*). Furthermore, the level of phospho-Mad in these discs is not changed (*Figure 2T*). These results indicate that in the wing disc, Babo-A activates dSmad2, not Mad, to regulate normal tissue growth.

Taken together, and assuming no difference in the number of wing disc precursors after embryogenesis, we infer that *babo-a* regulates tissue size by regulating both cell size and likely cell proliferation over the course of larval development. The data indicate that Babo-A is the only functional isoform in the wing imaginal disc and its removal phenocopies *myo* mutants in regulating wing disc growth. Thus, similar to the mushroom body neurons (*Awasaki et al., 2011*), our data demonstrate that Myo signals solely through Babo-A in the wing imaginal disc.

## Babo-A and Punt mediate non-canonical signaling in wing imaginal discs

In addition to canonical signaling, Babo can, under special circumstances, also signal non-canonically by cross-talking with downstream components of the BMP/Dpp pathway (*Peterson and O'Connor, 2013*). Non-canonical Babo activity is only detectable in a protein null mutant of *dSmad2* or a strong RNAi knockdown of *dSmad2* (*Peterson and O'Connor, 2013*; *Peterson et al., 2012*; *Sander et al., 2010*). In the absence of its preferred target dSmad2, Babo promiscuously interacts with Mad, the BMP ortholog of dSmad2, which then represses *brk* expression causing disc overgrowth along the anterior-posterior axis (*Peterson and O'Connor, 2013*). We hypothesized that in the absence of *dSmad2*, non-canonical signaling (i.e. silencing of *brk*) is also mediated by Babo-A and not through ectopic expression of other receptor isoforms. Since Babo activity in *dSmad2* loss-of-function RNAi clones results in silencing of the *brk* (*B14-LacZ*) reporter (*Müller et al., 2003*; *Peterson and O'Connor, 2013*), we used this assay to first generate *dSmad2* RNAi clones and quantified the ectopic *brk-LacZ* silencing (*Figure 3A, J*). Next, we performed genetic epistasis analysis by simultaneously knocking down *dSmad2* in clones along with each of the *babo* isoforms and then assayed for *brk-LacZ* rescue. Knocking down *dSmad2* and *babo-a* rescued *brk* reporter expression (*Figure 3*, *A vs B*), while RNAi of *dSmad2* and *babo-b* or *babo-c* did not (*Figure 3*, *A vs C-D*).

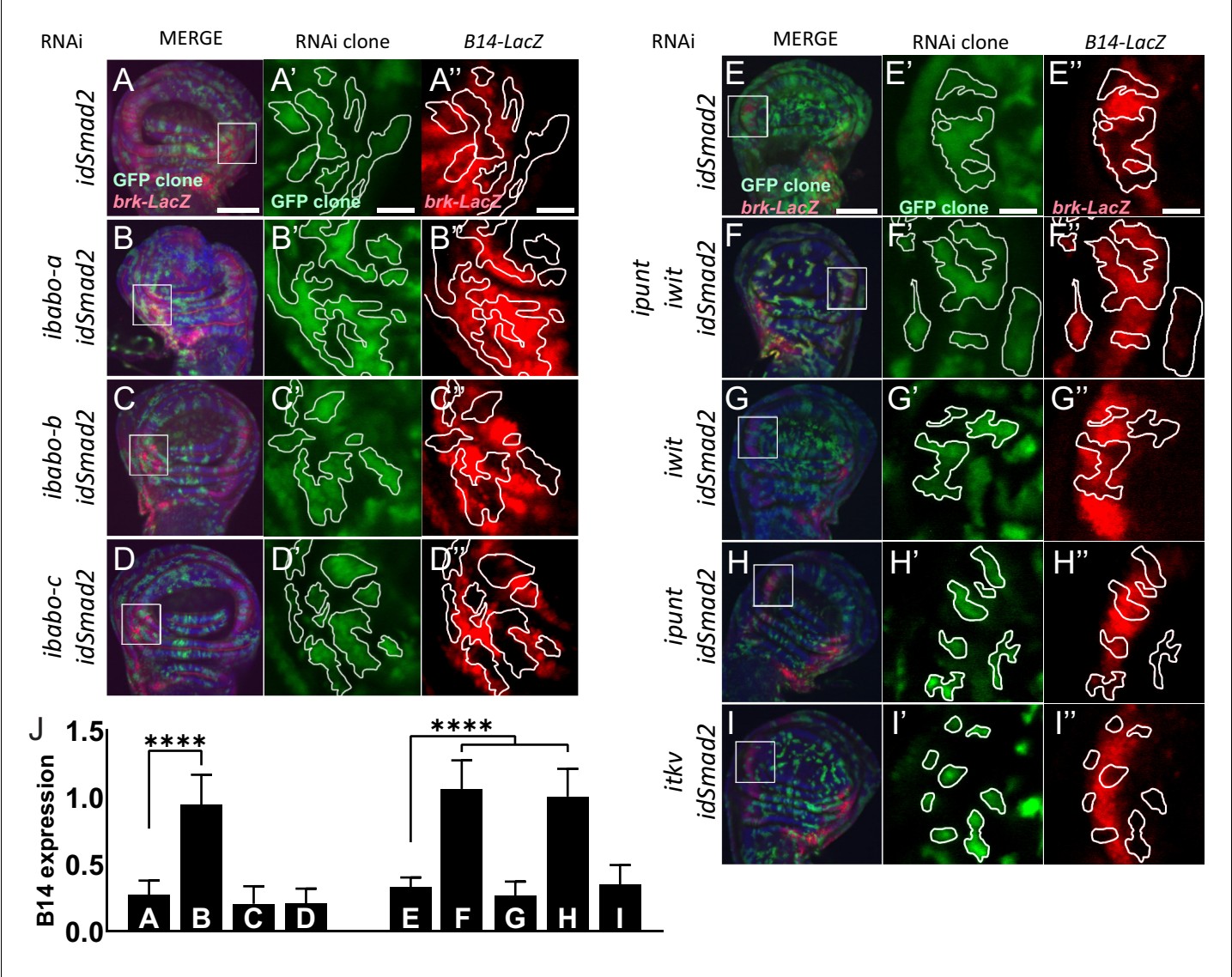

**Figure 3.** Babo-A and Punt are required for non-canonical silencing of *brk*. In the background of *brk-LacZ* reporter line random GFP+ (green) clones were induced by heat shock 48 hr prior to analysis (see methods for details). GFP+ marked clones also express RNAi for the indicated genes to the left of respective panels. Clones at the lateral edges of the disc were analyzed for patterning defects of the *brk* reporter. (A-D) Analysis of Babo isoforms. (E-I) Analysis of Type-II receptors. (A-I) Maximum intensity projections of wing imaginal discs stained with DAPI (blue) and anti β-gal (red), and fluorescent clone marker (green), scale bar = 100 µm. (A'-I') Single confocal section of higher magnification of insert in (A-I) showing GFP+ marked clone expressing RNAi for indicated genes. Clones are outlined with white lines, scale bar = 25 µm. (A''-I'') Same confocal section as in (A'-I') with outlined clones showing *brk-LacZ* reporter expression pattern, scale bar = 25 µm. *Brk* reporter is ectopically silenced in dSmad2 RNAi clones (A and E). This ectopic silencing is rescued by the concomitant RNAi of *babo-a* alone (A'' vs B'') but not *babo-b* or *babo-c* (A'' vs C'' or D''). dSmad2 RNAi dependent *brk* reporter silencing (E'') can be rescued by the concomitant RNAi of both Type-II receptors *punt* and *wit* (E'' vs F''). RNAi for *wit* does not rescue *brk* reporter (E'' vs G''). RNAi for *punt* completely rescues the *brk* reporter (E'' vs H''). *brk* silencing in *dSmad2* RNAi clones at the lateral edge of the disc is not mediated by *thickveins* (*tkv*, BMP Type-I receptor), since concomitant RNAi for *tkv* does not rescue ectopic *brk* silencing (E'' vs I''), indicating that *brk* silencing in *dSmad2* RNAi clones occurs via activation of TGFβ/Activin signaling. (J) Quantification of relative *brk-LacZ* reporter expression in GFP+ clones (n = 9 clones per condition).

The online version of this article includes the following figure supplement(s) for figure 3:

**Figure supplement 1.** RNAi for *babo-a* rescues wing disc overgrowth due to *dSmad2* RNAi.

To further test if non-canonical activity of Babo-A occurs throughout the wing disc, we knocked down *dSmad2* in the entire disc using *esg-GAL4*, resulting in disc overgrowth which can be quantified by measuring the width/height ratio of the disc-pouch (*Figure 3—figure supplement 1, A–B*; *Peterson and O'Connor, 2013*; *Sander et al., 2010*). RNAi for *dSmad2* alone results in the W/H ratio increasing from 1.1 (ctrl, *esg>*) to 1.5 (*esg >idSmad2*) (*Figure 3—figure supplement 1, D–E*). Similar to the *B14* de-repression, epistasis analysis with specific *babo* isoforms demonstrates that simultaneous RNAi of *babo-a*, and not *babo-b* or *babo-c*, suppresses the widening induced by *dSmad2* RNAi (*Figure 3—figure supplement 1*). Taken together, these experiments suggest that Babo-A is the sole signaling receptor in the wing imaginal disc mediating both canonical and non-canonical signaling.

TGFβ signaling requires ligands binding to hetero-tetrameric type-I and type-II receptor complexes. There are two type-II receptors in the *Drosophila* genome: *punt* and *wit*, both of which can signal with Type-I receptors from either branch of the TGFβ superfamily (*Upadhyay et al., 2017*). Punt is more commonly utilized during development, and facilitates Dpp signaling in the wing imaginal disc in combination with the BMP type I receptors Tkv and Sax (*Brummel et al., 1994*; *Fuentes-Medel et al., 2012*; *Letsou et al., 1995*; *Parker et al., 2006*). In cell culture models however, Punt can also signal in combination with Babo and Activin-like ligands (*Brummel et al., 1999*; *Jensen et al., 2009*; *Letsou et al., 1995*). Conversely, Wit functions primarily at the neuromuscular junction with the BMP ligand Gbb (*Marqués et al., 2002*) although it has been shown to bind Myo in combination with Babo in mammalian cell culture (*Lee-Hoeflich et al., 2005*). Since both *punt* and *wit* are expressed in the wing imaginal disc, it is unclear whether one or both type II receptors are required for Myo-Babo signaling (*Childs et al., 1993*; *Marqués et al., 2002*). To address this issue, we utilized the *brk* silencing assay in RNAi clones as described above. First, we find that knocking down both *punt* and *wit* along with *dSmad2* completely rescues the ectopic *brk* silencing (*Figure 3*, E vs F). This result indicates that one or both of these receptors is required for *brk* silencing in the *dSmad2* RNAi clones. Next, we find that RNAi for *wit* does not rescue *brk* silencing (*Figure 3*, E vs G), however RNAi for *punt* completely suppresses the *dSmad2* RNAi induced *brk* silencing (*Figure 3*, E vs H). This result demonstrates that Punt signaling occurs, however it does not rule out that it may do so through Tkv, the Dpp type I receptor. To exam this issue, we knocked down *tkv* and *dSmad2* in clones and looked for *brk* silencing. We find that *dSmad2* RNAi silenced *brk* even in the absence of *tkv* demonstrating that Dpp and Tkv signaling is not required for this phenotype and that Babo is the primary type I receptor mediating these clonal effects. Taken together, these experiments demonstrate that Babo-A requires Punt, but not Wit, for repression of *brk* expression in this non-canonical signaling assay and it is likely that the same receptors are used for Myo signaling in wild type disc cells.

## Muscle-derived Myoglianin signals to the wing disc

Growth factors such as Wg and Dpp are tissue intrinsic ligands, produced by and acting on the wing disc cells. We sought to determine if the same is true for Myo. The name *myoglianin* refers to the two tissues (muscles and glia) in which *myo* expression was initially observed (*Lo and Frasch, 1999*). More recent RNA in situ hybridization results suggested that *myo* is also expressed in the wing imaginal disc at low levels (*Hevia and de Celis, 2013*). To precisely determine the source of Myo that acts on the wing disc, we first examined the expression pattern of *myo* in late L3 larvae using a *myo-GAL4* reporter to drive *UAS-CD8-GFP* (*Awasaki et al., 2011*). We detect GFP expression in both Repo+ glial cells of the brain and the ventral ganglia, as well as the larval body wall muscles (*Figure 4A-B*, *Figure 4—figure supplement 1*), but not in any imaginal discs (*Figure 4A*, *Figure 4—figure supplement 1*). Thus, we conclude that either *myo* is not expressed in the wing disc or that our *myo-GAL4* construct does not include all of the endogenous enhancers.

To determine the functional source of Myo that controls imaginal disc growth, we conducted loss-of-function studies in the *dSmad2* null mutant background, and looked for suppression of the wing disc overgrowth phenotype. First, using null mutant alleles, we confirmed that loss of *myo* in *dSmad2* mutant background suppresses the *dSmad2* overgrowth, resulting in small discs similar in size to *myo* mutants (*Figure 4C-E, L*). Next, we knocked down *myo* using tissue specific GAL4 drivers. To confirm efficacy of the RNAi constructs, we first used the *da-GAL4* driver to ubiquitously knock down *myo* in the *dSmad2* background and confirmed that it suppresses wing disc overgrowth (*Figure 4*, F vs G, L). However knocking down *myo* using either *esg-GAL4* (wing disc) or *repo-GAL4*

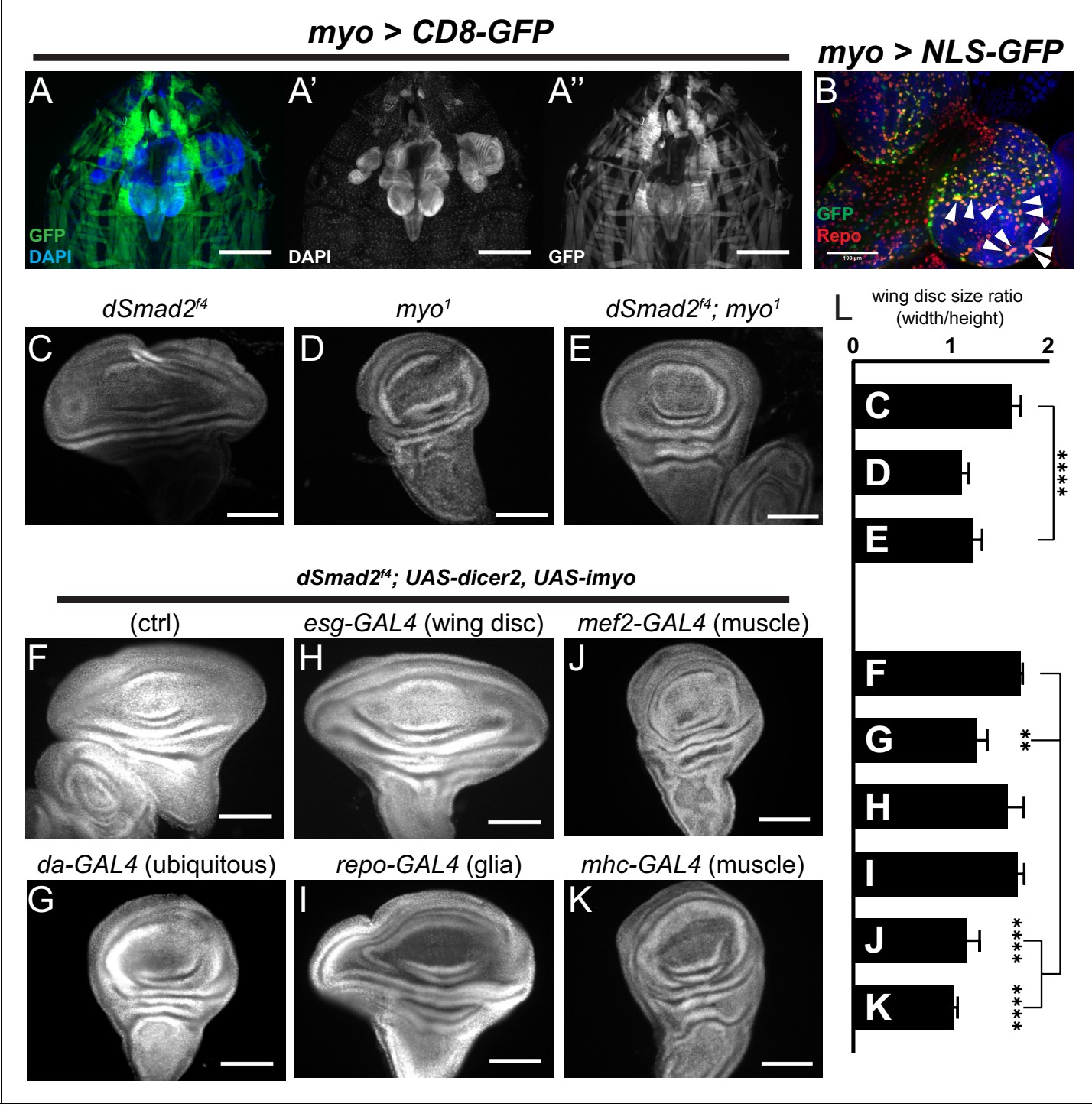

**Figure 4.** Muscle-derived Myo is required for non-canonical Babo activity in imaginal discs. (**A**) 3rd instar larval fillet showing *myo-GAL4* driving mCD8-GFP (green), DAPI (blue) counterstain to image all tissues in the field of view. DAPI channel (**A'**) shows muscle and cuticle-associated cell nuclei, as well as the brain and several imaginal discs. GFP channel (**A''**) represents the expression pattern of the *myo-GAL4* reporter transgene. Note that the wing imaginal discs does not express detectable GFP, scale bar = 500 µm. (**B**) Higher magnification of 3rd instar larval brain from *myo >NLS* GFP, and co-stained with anti-Repo marking glial cells. Arrowheads indicate overlap of *myo-GAL4* expression in a subset of Repo+ glial cells, scale bar = 100 µm. (**C–E**) *myo* is epistatic to *dSmad2*, and rescues the disc overgrowth phenotype, scale bar = 100 µm, n > 5. *dSmad2f4* mutant discs (**C**) are overgrown, however *myo* mutant discs (**D**) retain the normal shape of the tissue. The *dSmad2f4;myo* double mutant discs (**E**) are similar to *myo* mutant discs, demonstrating rescue of the overgrowth phenotype, and that Myo can function in non-canonical Babo activation. (**F–K**) Muscle specific RNAi of *myo* is sufficient to rescue wing disc overgrowth phenotype of *dSmad2* mutants, scale bar = 100 µm, n > 5 per group. RNAi of *myo* ubiquitously using *da-*

*Figure 4 continued on next page*

*Figure 4 continued*

GAL4 rescues the wing disc overgrowth (F vs G), phenocopying double *dSmad2;myo* double mutants (E vs G). RNAi of *myo* in either wing discs (*esg-GAL4*) or glia (*repo-GAL4*) is unable to rescue disc overgrowth (F vs H or I). RNAi of *myo* in larval muscles using *mef2-GAL4* or *MHC-GAL4* rescues the disc overgrowth phenotype (F vs J or K), which phenocopies ubiquitous *myo* RNAi (G vs J or K). (L) Quantification of wing shape widening in C-K. *dSmad2* wide wing is rescued by *myo* (C vs E) and knockdown of *myo* in muscles (F vs J or K).

The online version of this article includes the following figure supplement(s) for figure 4:

**Figure supplement 1.** Higher magnification of larval filet showing *myo-GAL4* expression pattern.

(glia) failed to suppress the *dSmad2* wing disc overgrowth (*Figure 4*, *F vs H, I*). These results suggest that the wing discs are receiving a strong Myo signal even when *myo* is knocked down in the imaginal discs or glia. In contrast, RNAi knockdown of *myo* in body wall muscles, using either *mhc-GAL4* or *mef2-GAL4* drivers, completely suppresses the *dSmad2* disc overgrowth phenotype phenocopying the *dSmad2;myo* double mutant (*Figure 4*, *F vs J, K*). Therefore, we conclude that muscles are the major source of Myo that stimulates imaginal disc growth.

## Overexpression of Myo in muscle or fat body rescues *myo* mutant disc growth

While the suppression of disc overgrowth in a *dSmad2* mutant background strongly implicates muscle as the source of ligand, we also examined if Myo expression in muscle, or other non-disc tissues, could rescue the size of *myo* mutant discs. First, overexpression of *myo* in larval muscles in wild type animals results in wings that are larger than normal (*Figure 5—figure supplement 1*). Next, we examined whether overexpression of *myo* in glia, muscles, or fat body is able to rescue the size of *myo* mutant discs. Overexpression in glia (*repo-GAL4*) results in partial rescue (*Figure 5C and F*) while overexpression in muscle (*mef2-GAL4*, *Figure 5D and F*) or fat body (*cg-GAL4*, *Figure 5E and F*) leads to complete rescue. These results clearly demonstrate that Myo produced in distant tissues can signal to wing discs, albeit with varying degrees of signaling strength depending on the source. Collectively, these results indicate that muscle-derived Myo acts as an endocrine-like ligand via the hemolymph to activate a specific receptor complex on disc cells, consisting of Babo-A and Punt, that signals to dSmad2 to promote imaginal disc growth.

## Discussion

Understanding the physiologic roles of inter-organ signaling during development is an emerging and complex field. In particular, how and which inter-organ signals regulate organ/tissue growth remains an open question. In the present study, we demonstrate that the *Drosophila* Activin-like ligand Myoglianin is a myokine that acts systemically to fine tune imaginal disc growth. We suggest Myo tuning of disc growth may play an important role in maintaining proper allometric scaling of appendage size with the size of the muscles that control their movements.

### Only Myo regulates imaginal disc size

Although Babo/dSmad2 signaling has been previously implicated in imaginal disc growth control (*Brummel et al., 1999*; *Hevia and de Celis, 2013*; *Peterson and O'Connor, 2013*), the ligand(s) responsible and their production sites(s) have not been identified. Previous *in situ* hybridization and RNAi knockdown experiments suggested that all three Activin-like ligands (Myoglianin, Activinβ, and Dawdle) contribute to control of wing size (*Hevia and de Celis, 2013*). However, we find no expression of these Activin-like ligands in imaginal discs, with the exception of *Actβ* which is expressed in differentiating photoreceptors of the eye imaginal disc (*Moss-Taylor et al., 2019*; *Ting et al., 2007*; *Zhu et al., 2008*). More importantly, using genetic null mutants, we show that only loss of *myo* affects imaginal disc size. The discrepancy in phenotypes between tissue-specific knockdown results and the genetic nulls is often noted and not fully understood (*Di Cara and King-Jones, 2016*; *Gibbens et al., 2011*). In addition to simple off-target effects within the wing disc itself, one possible explanation is that many GAL4 drivers are expressed in tissues other than those reported, potentially resulting in deleterious effects for the animal that indirectly affect imaginal disc size. Another possibility is that in *Actβ* and *daw* genetic null backgrounds a non-autonomous compensatory signal is

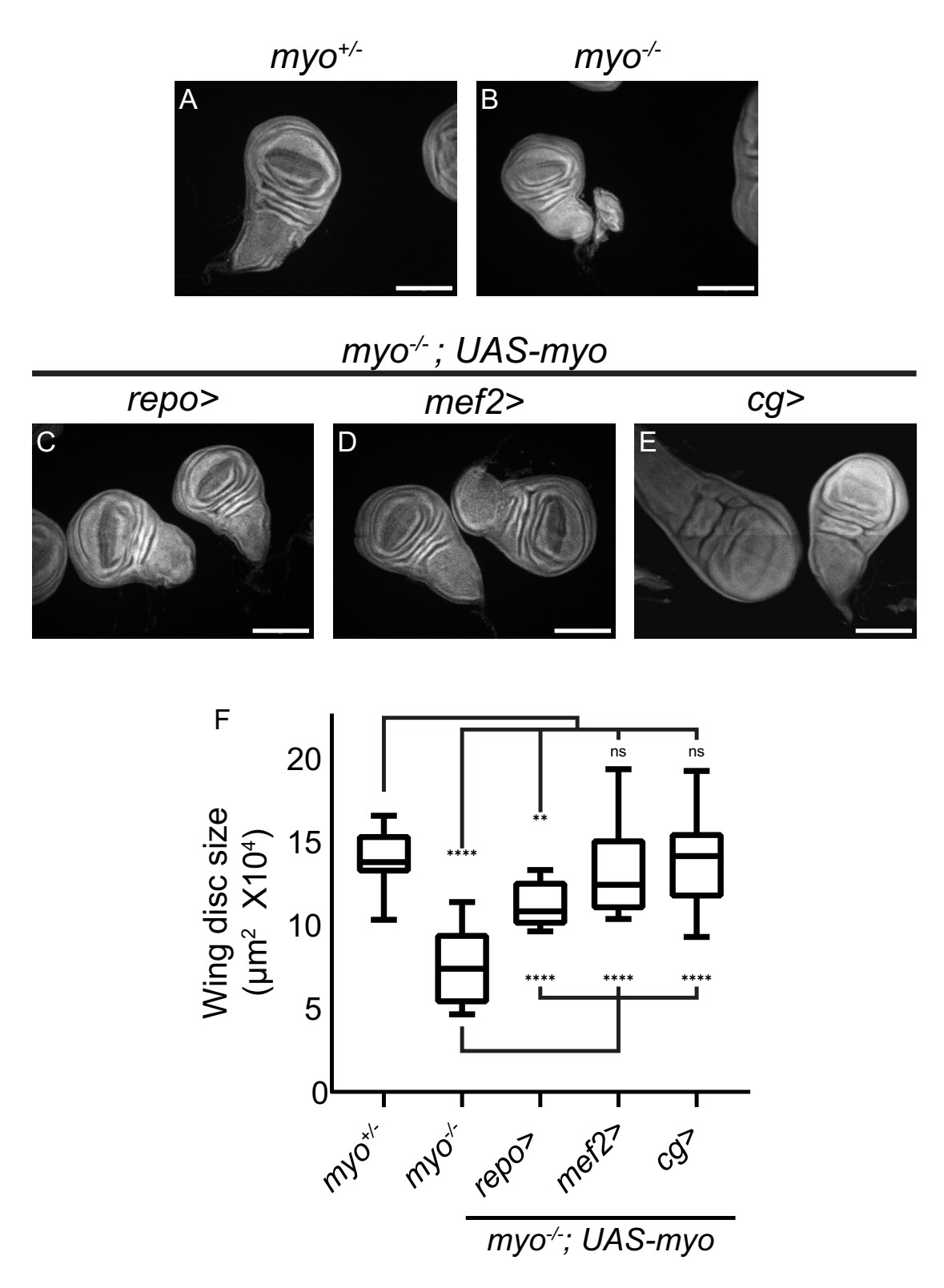

**Figure 5.** Overexpression of *myo* in muscles or fat body rescues wing disc size of *myo* mutants. (A–B) *Myo* mutant discs (B) are smaller than heterozygous controls (A), and (C–E) wing discs from *myo* overexpression rescue experiments, scale bar = 100 μm, n > 19. *Myo* overexpression in glial cells using *Repo-GAL4* partially rescues wing disc size (B vs C). Overexpression in muscles completely rescued *myo* mutant discs back to wild type size
*Figure 5 continued on next page*

*Figure 5 continued*

(B vs D vs A). *Myo* is not normally expressed in the fat body (*Figure 4*). However, ectopic overexpression in the fat body was sufficient to rescue discs back to wild-type size (B vs E vs A). (F) Quantification of wing disc size from (A–E).

The online version of this article includes the following figure supplement(s) for figure 5:

**Figure supplement 1.** Ectopic Myo expression in muscle produces larger wings.

generated by another tissue and this signal is not activated in the case of tissue-specific knockdown (*Di Cara and King-Jones, 2016*). We think both of these explanations are unlikely in this instance since we demonstrate that only the Babo-A receptor isoform is expressed and required in discs. Since we have previously shown that Daw only signals through isoform Babo-C, it is unclear why knockdown of *daw* in the wing disc would result in a small wing phenotype as previously reported (*Hevia and de Celis, 2013*; *Jensen et al., 2009*). We conclude that the small wing phenotypes caused by RNAi knockdown of *Actβ* or *daw* are likely the result of off-target effects and that Myo is the only Activin-type ligand that regulates imaginal disc growth.

## Punt is required for efficient Babo-A signaling to dSmad2 in the wing disc

The signaling ability of TGFβ ligands is modulated by the specific combinations of receptors and co-receptors to which they bind (*Heldin and Moustakas, 2016*). In *Drosophila*, the receptor requirements for effective signaling through dSmad2 likely vary for each ligand and tissue (*Jensen et al., 2009*). In this report, we find that Myo signaling in the wing disc requires Punt as the type II receptor and Babo-A as the type I receptor. Furthermore, we establish that Myo is the exclusive Activin-like ligand signaling to the discs since loss of Myo eliminated detectable phosphorylation of dSmad2 in the wing imaginal disc. Since Babo-A is the only isoform expressed in wing discs, we also conclude that Myo is able to signal through this isoform in the absence of other isoforms. Whether Myo can also signal through Babo-B or C is not yet clear, but in the context of mushroom body remodeling Babo-A also appears to be the major receptor isoform utilized (*Awasaki et al., 2011*; *Yu et al., 2013*). The co-receptor Plum is also required for mushroom body remodeling, suggesting that Plum and Babo-A are both necessary for efficient Myo signaling. However, it is noteworthy that Plum null mutants are viable (*Yu et al., 2013*) while Myo null mutants are not. This observation suggests that Plum is not required for all Myo signaling during development. Further studies will be required to evaluate whether Plum is essential to mediate Myo signaling in imaginal discs.

The requirement of Punt as a type II receptor for production of an efficient signaling complex with Myo may be context dependent. In the mushroom body, indirect genetic evidence suggests that the two Type II receptors function redundantly (*Zheng et al., 2006*). Although both *punt* and *wit* are expressed in imaginal discs (*Childs et al., 1993*; *Marqués et al., 2002*), only loss of *punt* produces a phenotype in the *brk* reporter assay. To date, we have not been able to see clear signaling in S2 cells expressing Punt and Babo-A when Myo is added. It is also notable that a previous attempt to study Myo signaling in a heterotypic cell culture model also failed (*Lee-Hoeflich et al., 2005*). In that study, Myo was found to form a complex with Wit and Babo-A in COS-1 cells but no phosphorylation of dSmad2 was reported. One explanation is that effective signaling by Myo requires Punt, and Babo-A, and perhaps another unknown co-receptor that substitutes for Plum. Despite this caveat, our results provide in vivo functional evidence for a Myo signaling complex that requires Babo-A and Punt to phosphorylate dSmad2 for regulation of imaginal disc growth.

## Proliferation and cell size

Final tissue size is determined by several factors including cell size, proliferation, death rates, and duration of the growth period. While we did observe cell size changes upon manipulation of Myo signaling, the direction of change depended on the genotype. In *myo* mutants, estimation of cell size via apical surface area indicates that the cells are ~20% smaller than wild-type. Although this measurement does not indicate the actual volume of the cells, it gives an indication of cell density in the epithelial sheet of the wing pouch, which is analogous to counting cells in the adult wing. RNAi knock down of *babo-a* in the entire disc produced smaller adult wings with larger (less dense) cells. This result differs from the *myo* mutant, but is similar to the reported adult wing phenotypes of

*babo* mutants (*Hevia and de Celis, 2013*) and larval disc phenotypes of *dSmad2* mutants (*Peterson and O'Connor, 2013*). When *babo-a* is knocked down in one compartment, that compartment is reduced in size with smaller cells. We conclude that tissue size reduction is the consistent phenotype upon loss of Myo signaling, but cell size changes depend on the specific type of manipulation.

While cell size effects may be context dependent, it is notable that neither reduction in size of imaginal discs nor adult wing surface area can be explained solely by a cell size defect. Since we see no apoptotic increase in *myo* mutant discs, and because *dSmad2* knockdown also fails to alter apoptotic rate (*Hevia and de Celis, 2013*), the mostly likely cause is an altered proliferation rate. Consistent with this view is that the large disc phenotype exhibited by dSmad2 protein null mutants is clearly dependent on Myo and we and others have previously shown that this is the result of enhanced proliferation (*Peterson and O'Connor, 2013*; *Hevia and de Celis, 2013*). Similarly earlier studies also showed that expression of activated Babo or activated dSmad2 in wing discs also leads to larger wings with slightly smaller cells which is most easily explained by an enhanced proliferation rate (*Brummel et al., 1999*; *Hevia and de Celis, 2013*). It is worth noting that this proposed enhanced proliferation rate is difficult to detect since cell division is random with regard to space and time during development (*Wartlick et al., 2011*). Thus a ~ 20% reduction in adult wing size caused by a proliferation defect translates into about 1/5[th] of disc cells dividing on average one less time throughout the entire time course of larval development. Therefore, without prolonged live imaging, this small reduction in proliferation rate will not be detectable using assays that provide only static snapshots of cell division. It is worth noting, however, that previous clonal studies also concluded that dSmad2 or Babo loss in wing disc clones resulted in a reduced proliferation rate (*Hevia and de Celis, 2013*).

One attempt to shed light on the transcriptional output of TGFβ signaling responsible for wing disc size employed microarray mRNA profiling of wild-type versus *dSmad2* gain- and loss-of-function wing discs (*Hevia et al., 2017*). However, this study did not reveal a clear effect on any class of genes including cell cycle components, and it was concluded that the size defect is the result of small expression changes of many genes. Consistent with this view are dSmad2 Chromatin Immunoprecipitation experiments in Kc cells which revealed that dSmad2 is associated with many genomic sites and thus may regulate a myriad of genes (*Van Bortle et al., 2015*).

## Is *myoglianin* function evolutionarily conserved?

Insect *myoglianin* is a clear homolog of vertebrate *Myostatin* (*Mstn/GDF8*) (*Hinck et al., 2016*), a TGFβ family member notable for its role in regulating skeletal muscle mass. *Mstn* loss-of-function mutants lead to enlarged skeletal muscles (*Grobet et al., 1997*; *McPherron and Lee, 1997*). Mstn is thought to affect muscle size through autocrine signaling that limits muscle stem cell proliferation, as well as perturbing protein homeostasis via the Insulin/mTOR signaling pathways (*Morikawa et al., 2016*; *Rodriguez et al., 2014*). Similarly, *Gdf11*, a *Mstn* paralog, also regulates size and proliferation of muscles and adipocytes, and may promote healthy aging (*Egerman et al., 2015*; *McPherron et al., 2009*; *Sinha et al., 2014*). *Mstn* and *Gdf11* differ in where they are expressed and function. *Mstn* is highly expressed in muscles during development while *Gdf11* is weakly expressed in many tissues. Both molecules are found to circulate in the blood as latent complexes in which their N-terminal prodomains remain associated with the ligand domain. Activation requires additional proteolysis of the N-terminal fragment by Tolloid-like metalloproteases to release the mature ligand for binding to its receptors (*Uhlén et al., 2015*; *Wolfman et al., 2003*). Interestingly in *Drosophila*, the Myoglianin N-terminal domain was also found to be processed by Tolloid-like factors, but whether this is a prerequisite for signaling has not yet been established (*Serpe and O'Connor, 2006*). In terms of functional conservation in muscle size control, our results of both null mutants and RNAi depletion indicates that it has little effect on muscle size. This contradicts a previous study in which muscle-specific RNAi knockdown of *myo* was reported to produce larger muscles similar to the vertebrate observation (*Augustin et al., 2017*). The discrepancy between our tissue-specific RNAi knockdown and previous studies is not clear, but our null mutant analysis strongly argues that *Drosophila* Myo does not play a role in muscle size control. Intriguingly however, we find that loss of *Actβ*, another ligand that signals through Babo and dSmad2, results in a smaller muscles (*Moss-Taylor et al., 2019*) contrary to that produced by loss of vertebrate *Mstn* and various other vertebrate Activin family members. Recent data has shown that *DrosophilaActβ* is the only Activin-

like ligand that affects muscle growth, and it does so, in part, by regulating Insulin/Tor signaling in the opposite direction compared to vertebrates (*Kim and O'Connor, 2020*). Thus, in *Drosophila* the Myo/Activin pathway promotes muscle growth while in vertebrates it inhibits muscle growth.

## Intrinsic (intra-organ) vs extrinsic (inter-organ) mode of Myo signaling and the significance of muscle to disc inter-organ communication

The most intriguing finding of this study is that muscle-derived Myo acts non-autonomously to regulate imaginal disc growth. This is in stark contrast to the two BMP ligands, Dpp and Gbb, which are produced by disc cells and act autonomously within the disc itself to regulate both growth and pattern (*Haerry et al., 1998*; *Minami et al., 1999*). The fact that a TGFβ ligand can act in an endocrine-like manner is not particularly novel since many vertebrate members of the TGFβ family, including Myostatin, the closest homolog to *Drosophila* Myoglianin, are found in the blood (*Sinha et al., 2014*). Even the disc intrinsic molecule Dpp has been recently shown to be secreted into the hemolymph where it circulates and signals to the prothoracic gland to regulate a larval nutritional checkpoint (*Setiawan et al., 2018*). Several additional reports indicate that ligands from the *Drosophila* Activin-like subfamily also circulate in the hemolymph and function as inter-organ signals. For example, muscle-derived Actβ and Myo signal to the fat body to regulate mitochondrial function and ribosomal biogenesis, respectively (*Demontis et al., 2014*; *Song et al., 2017*). In addition, Daw produced from many tissue sources can signal to the Insulin producing cells and the midgut to stimulate Insulin secretion and repress expression of sugar metabolizing genes, respectively (*Chng et al., 2014*; *Ghosh and O'Connor, 2014*). Thus, many TGFβ type factors act as both paracrine and endocrine signals depending on the tissue and process involved.

The phenotype of the *myo* mutant animal supports our claim that endogenous Myo contributes to imaginal disc growth. The ectopic expression assay produced various wing disc sizes when Myo was expressed in different tissues, indicating that the growth response likely depends on the level of Myo being produced in the distal tissue. Loss of glial derived Myo is not sufficient to suppress overgrowth of *dSmad2* mutant discs, but overexpression of Myo in glia did partially rescue size of *myo* null wing discs, likely because the *repo*-Gal4 driven overexpression produces more ligand than endogenous glia. Likewise, expression from a large tissue such as muscle or fat body likely produces more Myo than glia leading to normal disc growth or even overgrowth. It is also possible that Myo signaling activity is modified depending on the tissue source. Like other TGFβ family members, Myo requires cleavage by a furin protease at its maturation site to separate the C-terminal ligand from the prodomain (*Lee-Hoeflich et al., 2005*; *Lo and Frasch, 1999*). Myo may also require a second cleavage by a Tolloid protease family member to achieve full dissociation of the prodomain from the ligand to ensure complete activation (*Serpe and O'Connor, 2006*). Either of these cleavage reactions, or any other step impacting the bioavailability of active Myo ligand, may vary with tissue or may be modulated by environmental conditions.

What is the rationale for larval muscle regulating imaginal discs size? A possible reason is that for proper appendage function, the muscle and the structure (leg, wing, and haltere) that it controls should be appropriately matched to ensure optimal organismal fitness for the environmental niche the adult occupies. For example, a large muscle powering a small wing might result in diminished fine motor control. Conversely, a small muscle may not be able to power a large wing to support flight. However, the multi-staged nature of muscle and appendage development complicates this picture. Larval muscles are histolysed during metamorphosis and do not contribute to the adult muscle. However, remnants of larval muscles in the thoracic segment are preserved as fibers that act as scaffolds upon which the larval myoblasts infiltrate and fuse to become the adult indirect flight muscles (*Dutta et al., 2004*). Thus, at least for the indirect flight muscles, the size of the larval muscle scaffold might contribute to the building of a bigger adult muscle. Another possibility invokes a signal relay system. Wg and Serrate/Notch signaling from the wing disc epithelial cells control myoblast proliferation during larval development (*Gunage et al., 2014*). Thus it may be that Myo signaling from the larval muscles stimulates proliferation of the disc epithelial layer which in turn enhances Wnt and Serrate/Notch signaling to myoblasts to increase their number thereby coordinating the adult appendage size with muscle size. A final scenario is that, since muscle is a major metabolic and endocrine organ, Myo production may be regulated by the general metabolic state of the larva. If healthy, high levels of Myo, in concert with other growth signals such as insulin, leads to a bigger fly

with large wings, and if the metabolic state is poor then lower Myo levels leads to diminished proliferation and a smaller cell size resulting in a smaller fly with small wings.

Regardless of the precise mechanism, the ability of the muscle to control appendage size has interesting implications in terms of evolutionary plasticity. The proportionality of insect wing size to body size can vary over a large range (*Shingleton et al., 2008*), but the mechanism responsible for determining this particular allometric relationship for a given species is not understood. We have recently demonstrated that in *Drosophila*, motor neuron derived Actβ, another TGFβ superfamily member, can dramatically affect muscle/body size (*Moss-Taylor et al., 2019*). Therefore, it is tempting to speculate that evolutionary forces might modulate the activity of these two genes to produce an appropriate body-wing allometry that is optimal for that species' ecological niche.

# Materials and methods

## Key resources table

| Reagent type (species) or resource | Designation | Source or reference | Identifiers | Additional information |
|---|---|---|---|---|
| Genetic reagent (*D. melanogaster*) | w[1118] | Bloomington *Drosophila* Stock Center | RRID:BDSC_5905 | |
| Genetic reagent (*D. melanogaster*) | myo[1] | *Awasaki et al., 2011* | FLYB:FBal0267082 | Synonym: myo[Δ1] |
| Genetic reagent (*D. melanogaster*) | Actβ[80] | *Zhu et al., 2008* | FLYB:FBal0217392 | Synonym: Actβ[ed80] |
| Genetic reagent (*D. melanogaster*) | daw[11] | *Serpe and O'Connor, 2006* | FLYB:FBal0212326 | |
| Genetic reagent (*D. melanogaster*) | myo[CR2] | This paper | | See Materials and methods |
| Genetic reagent (*D. melanogaster*) | dSmad2[F4] | *Peterson et al., 2012* | FLYB:FBal0268512 | Synonym: Smox[F4] |
| Genetic reagent (*D. melanogaster*) | babo-a miRNA | *Awasaki et al., 2011* | FLYB:FBtp0071178 | Also a new allele at VK37; see Materials and methods |
| Genetic reagent (*D. melanogaster*) | babo-b miRNA | *Awasaki et al., 2011* | FLYB:FBtp0071179 | |
| Genetic reagent (*D. melanogaster*) | babo-c miRNA | *Awasaki et al., 2011* | FLYB:FBtp0071180 | |
| Genetic reagent (*D. melanogaster*) | Myo-GAL4 | *Awasaki et al., 2011* | FLYB:FBtp0071181 | |
| Genetic reagent (*D. melanogaster*) | esg-GAL4 | Bloomington *Drosophila* Stock Center | FLYB:FBal0098823 | Lab stock is esg-GAL4/Cyo-GFP |
| Genetic reagent (*D. melanogaster*) | ci-GAL4 | Herman Steller | FLYB:FBtp0022318 | Source obtained from G Morata |
| Genetic reagent (*D. melanogaster*) | hh-GAL4 | Herman Steller | FLYB:FBal0121962 | Source obtained from G Morata |
| Genetic reagent (*D. melanogaster*) | da-GAL4 | Bloomington *Drosophila* Stock Center | FLYB:FBal0042573 | |
| Genetic reagent (*D. melanogaster*) | myo miRNA | *Awasaki et al., 2011* | FLYB:FBtp0071174 | |
| Genetic reagent (*D. melanogaster*) | Mef2-GAL4 | Bloomington *Drosophila* Stock Center | FLYB:FBal0052385 | |
| Genetic reagent (*D. melanogaster*) | idSmad2 | *Peterson et al., 2012* | FLYB:FBal0268514 | |
| Genetic reagent (*D. melanogaster*) | B14-LacZ | *Müller et al., 2003* | FLYB:FBal0148007 | |

*Continued on next page*

*Continued*

| Reagent type (species) or resource | Designation | Source or reference | Identifiers | Additional information |
|---|---|---|---|---|
| Genetic reagent (*D. melanogaster*) | iPunt | This paper | | See Materials and methods |
| Genetic reagent (*D. melanogaster*) | iWit | This paper | | See Materials and methods |
| Genetic reagent (*D. melanogaster*) | iTkv | *Peterson et al., 2012* | FLYB:FBal0268515 | |
| Genetic reagent (*D. melanogaster*) | iMyo | This paper | | See Materials and methods |
| Genetic reagent (*D. melanogaster*) | Repo-Gal4 | Bloomington *Drosophila* Stock Center | FLYB:FBal0127275 | |
| Genetic reagent (*D. melanogaster*) | Mhc-Gal4 | Bloomington *Drosophila* Stock Center | FLYB:FBti0160475 | P{Mhc-Gal4.K}2 |
| Genetic reagent (*D. melanogaster*) | Mhc(f)-GAL4 | N. Perrimon | FLYB:FBti0012514 | P{GAL4-Mhc.W}MHC-82 |
| Genetic reagent (*D. melanogaster*) | Cg-Gal4 | Bloomington *Drosophila* Stock Center | FLYB:FBal0104726 | |
| Genetic reagent (*D. melanogaster*) | UAS-Myo-7D3 | *Gesualdi and Haerry, 2007* | FLYB:FBtp0039710 | |
| Genetic reagent (*D. melanogaster*) | iRFP | Bloomington *Drosophila* Stock Center | BDSC:41554 | |
| Genetic reagent (*D. melanogaster*) | iGFP | Bloomington *Drosophila* Stock Center | BDSC:67852 | |
| Antibody | p-dSmad2 (rabbit monoclonal) | Cell Signaling Technology | Cat# 3108; RRID:AB_490941 | WB (1:1000); detects phosphorylated dSmad2 |
| Antibody | Tubulin (mouse monoclonal) | Sigma-Aldrich | Cat# T9026 | WB (1:1000) |
| Antibody | p-Mad (rabbit polyclonal) | *Peterson et al., 2012* | | WB (1:500); detects phosphorylated Mad |
| Antibody | Repo (mouse monoclonal) | DSHB | Cat# 8D12; RRID:AB_528448 | IF(1:100) |

## Fly strains

TGFβ mutants: *myo¹*, *Actβ⁸⁰*, *daw¹¹*, *dSmad2^F4* are described previously (*Awasaki et al., 2011*; *Peterson and O'Connor, 2013*; *Serpe and O'Connor, 2006*; *Zhu et al., 2008*). The new *myo^CR2* allele is described in *Kim and O'Connor, 2020*. Isoform miRNA lines for *babo* and *myo-GAL4* are described previously (*Awasaki et al., 2011*). We generated a new line of UAS-driven *babo-a* miRNA at the VK37 attP site using the published construct (used for *Figure 2S,T*). RNAi lines for *dSmad2* and *tkv* and the strategy used to generate new RNAi lines are described previously (*Peterson and O'Connor, 2013*; *Peterson et al., 2012*). New RNAi lines for *wit* targeted 1103–1608 (NM_079953.3), and *punt* targeted 1069–1582 (NM_169591.1). RNAi lines for *myo* targeted 2259–3503 (NM_166788.2) utilizing the 'snapback' gDNA/cDNA hybrid method (*Kalidas and Smith, 2002*). Stocks and methods used for induction of GAL4 flp-out clones are described previously (*Peterson and O'Connor, 2013*). Briefly, clones were induced by incubation at 37°C for 30 min. Larvae were dissected 48 hr later at wandering L3 stage.

*Ci-GAL4* and *hh-GAL4* were kind gifts from H. Steller (Rockefeller University). A version of MHC we labeled *MHC^f-GAL4* was a kind gift from N. Perrimon (HHMI, Harvard). The following reagents are available from Bloomington *Drosophila* Stock Center: *da-GAL4*, *esg-GAL4*, *repo-GAL4* (#7415),

*mef2-GAL4* (#27390), *MHC-GAL4* (#55133), cg-GAL4 (#7011), GFP knockdown (#41554), RFP knockdown (#67852), *UAS-mCD8::GFP*, *UAS-GFP.nls*.

## Microscopy and antibodies

Primary antibodies for IHC include cleaved Caspase-3 (Cell Signaling #9661), phospho-Histone3 (S10) (Sigma, H0412), and Repo (DSHB 8D12). For B14-LacZ reporter detection, wing discs were stained with anti-β-galactosidase (Promega, #Z378A). AlexaFluor 568 or 647 (Invitrogen) conjugated secondary antibodies were used when required. Larval muscle and wing disc epithelium actin-belt were stained with rhodamine-phalloidin. To measure wing disc size, tissues were stained with DAPI. To assess *brk* reporter expression in clones, the red IHC signal was compared within a GFP-marked clone and a neighboring patch of control cells. Background signal was subtracted, and a ratio was calculated such that one indicates no change of reporter signal in the clone and lower numbers indicated repression of *brk*. Clones from multiple discs were selected from three regions that normally express the reporter. Signals were extracted from single Z sections using a line plot (width 5) in FIJI and visualized with the RGBProfilesTools plug-in to estimate the clone boundary.

Tissues for fluorescence microscopy were mounted with 80% Glycerol in PBS (0.1% TritonX-100). Wide-field images taken with a 20X objective on a Zeiss Axiovert microscope. CARV Confocal images were captured using a 20X objective on a Zeiss Axiovert. We also used the Zeiss LSM710 for confocal and 'tile-scanning' imaging.

For adult wings, animals were dissected in 95% ethanol, and mounted with (50:50) Canadian balsam and wintergreen oil solution. Images were taken with a 4X objective (for whole wing) and 40X objective (for trichome density). Images were processed in fiji (imagej).

Wing disc size was calculated with the DAPI staining then using the threshold and measure functions. For larval muscle size and adult wing size, outlines were drawn using the polygon tool.

## Western blot

Mid-late third instar larvae were dissected in cold PBS, and wing imaginal discs were collected and washed on ice. Total protein was extracted with RIPA buffer (Cell Signaling) with cOmplete protease inhibitor (Roche) and PhosSTOP phosphatase inhibitor (Roche). Lysates were mixed with reducing sample buffer, boiled, and 10 µl of sample were run on 4–12% graded precast gels (Invitrogen). Resolved protein samples were transferred to the PVDF membrane. Membrane blocking and probing were performed using standard protocols for ECL. Antibodies used for western blot: phospho-Smad2 (Cell Signaling #3108), rabbit phospho-Smad3 polyclonal affinity purified with phospho-Mad peptide (*Peterson et al., 2012*), Tubulin (Sigma #T9026), goat anti-mouse HRP, and goat anti-rabbit HRP. phospho-dSmad2 were quantified in Fiji and normalized using tubulin or a non-specific band for loading control.

## qPCR

Late-third instar larvae were dissected in cold PBS, and tissues were pooled and washed in PBS on ice. Tissues were homogenized in Trizol (Invitrogen) and Total RNA purified using RNeasy Mini Kit (Qiagen). cDNA libraries were made using SuperScript-III (Invitrogen). qPCR was performed using SYBR green reagent (Roche) on a LightCycler 480. *Rpl23* was used as a housekeeping gene for normalization. We confirmed that primer sets for all three isoforms amplify equivalently using *babo-a, b* and *c* control cDNA templates. The following qPCR primers were used for *babo* isoforms: *babo-a*: F 5' GGCTTTTGTTTCACGTCCGTGGA 3' *babo-a*: R 5' CTGTTTCAAATATCCTCTTCATAATCTCAT 3' *babo-b*: F 5' GCAAGGACAGGGACTTCTG 3' *babo-b*: R 5' GGCACATAATCTTGGACGGAG 3' *babo-c*: F 5' GACCAGTTGCCACCTGAAGA 3' *babo-c*: R 5' TGGCACATAATCTGGTAGGACA 3'.

## Statistics

Statistical analyses were performed using Prism 8 (Graphpad). All experiments were repeated at least twice. Sample size was determined using published methods in the field, n values represent biological replicates per group. Data collection and analysis were not blind. Quantitative data are plotted as mean with standard deviations. Significance was determined using 2-tailed t-test with Welch's correction or an ANOVA with post-hoc multiple comparisons analysis. P values $< 0.05$ = *, $<0.01$ = **, $<0.001$ = ***.

## Acknowledgements

We would like to thank MaryJane O'Connor, Heidi Bretscher, Tom Neufeld, Jeff Simon, and Jim Ervasti for comments on the manuscript and valuable feedback on the project. Ashley Arthur conducted the experiment for *Figure 2—figure supplement 1*. We also thank Louise Lu for quantification of *babo* isoform primer amplification efficiency. Bloomington*Drosophila*Stock Center for numerous fly lines. *Ci-GAL4* and *hh-GAL4* lines were a gift from Herman Steller

## Additional information

### Funding

| Funder | Grant reference number | Author |
| --- | --- | --- |
| National Institute of General Medical Sciences | R35GM118029 | Michael B O'Connor |

The funders had no role in study design, data collection and interpretation, or the decision to submit the work for publication.

### Author contributions

Ambuj Upadhyay, Conceptualization, Data curation, Formal analysis, Validation, Investigation, Visualization, Methodology, Writing - editing; Aidan J Peterson, Conceptualization, Formal analysis, Investigation, Visualization, Methodology, Writing - review and editing; Myung-Jun Kim, Data curation, Formal analysis, Writing - review and editing, Dr. Kim made the myo Cr2 mutant that was used in Figure 1 S2 and provided the western blots shown in Fig 2 S and T; Michael B O'Connor, Conceptualization, Funding acquisition, Visualization, Project administration, Writing - review and editing

### Author ORCIDs

Aidan J Peterson http://orcid.org/0000-0002-6801-3364
Michael B O'Connor https://orcid.org/0000-0002-3067-5506

### Decision letter and Author response

Decision letter https://doi.org/10.7554/eLife.51710.sa1
Author response https://doi.org/10.7554/eLife.51710.sa2

## Additional files

### Supplementary files

• Transparent reporting form

### Data availability

All data generated or analysed during this study are included in the manuscript and supporting files.

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
