## [Decision Letter]

**Acceptance summary:**

We all enjoyed reading your paper and feel it is an important contribution to the field of developmental biology. The data that support that Muscle-derived Myoglianin regulates *Drosophilamelanogaster* imaginal disc growth is compelling and this will open new avenues of research. We also enjoyed reading your response, and as one reviewer stated, which we agree with, "this response and the sum of observations and possibilities the authors provided could be highlighted in the published response to reviewers." Thank you for submitting your work to *eLife*!

**Decision letter after peer review:**

Thank you for submitting your article "Muscle-derived Myoglianin regulates *Drosophilamelanogaster* imaginal disc growth" for consideration by *eLife*. Your article has been reviewed by three peer reviewers, and the evaluation has been overseen by a Reviewing Editor and Utpal Banerjee as the Senior Editor. The reviewers have opted to remain anonymous.

Reviewer #1:

The manuscript by Upadhyay et al. uses the imaginal disc/adult wing of *Drosophila* to examine molecular mechanisms that coordinate organ growth during development. The authors report that muscle secreted Myo controls the wing growth. They examine the components of the Myo signaling pathway and demonstrate that Babo-A, Punt and the co-receptor Plum are required for Myo-dependent regulation of the wing growth. Using elegant genetics and exploiting loss-of-function as well as gain-of-function phenotypes, the authors systematically built the case for each of the pathway components downstream Myo. They next turn towards the source of ligand and demonstrate that the productive Myo is secreted from the muscle. The manuscript is compelling and clearly written. The findings are interesting and timely, especially as more and more reports find that inter-organ signaling control multiple aspects of development. However, I do have a number of concerns that need to be addressed before this manuscript could be considered for publication.

First, the authors need to sort through some conflicting data published by other laboratories. However, they stop short of evaluating the reproducibility of some of these previous findings. Without turning this manuscript into a "set the record straight mission", I do think that this group of authors should use this stage to set the record straight and, in the process, validate some reagents or raise the appropriate flags. This process will benefit of the entire signaling community, especially since Myo has been the topic of extensive controversies.

For example, the authors write a Discussion section on Myo as being functionally distinct from vertebrate Mstn. While I agree with the authors' interpretation, as a reader I really want to know whether the *myo* null and the muscle knockdown of *myo* indeed show reproducible opposing muscle phenotypes; I want to be aware of such extreme off-targets or compensatory effects. The authors have already validated a *myo-RNAi* line and used it, including in the muscle (Figure 4). They should complete the comparison and include/discuss their data.

Secondly, Myo was previously shown to bind to Babo and Wit in mammalian tissue culture. Nonetheless, the authors demonstrate that Wit is dispensable for Myo signaling in the wing disc. Based on the non-canonical signaling analysis, the authors argue that Myo signals through Babo-A, Punt and Plum. The authors should cement this conclusion using signaling assays which could presumably recapitulate and illustrate both canonical and non-canonical signaling pathways downstream of Myo. Such results will tremendously strengthen the authors' proposed model and will increase the overall impact of the paper. Since, so far, it's been difficult to reconstitute signaling by *Drosophila* activin-type ligands in tissue culture, the field will be urged to take note of the co-receptor contributions.

Thirdly, the data presented here do not exclude an alternative explanation: That the positive effect of Myo onto the wing growth is due to both canonical and non-canonical signaling through Babo-A. The authors should include pMad analyses in Figure 1T to address this possibility. Also, they should provide pMad and pSmad2 panels for Figure 5—figure supplement 1.

Other concerns:

1) "muscle" is misspelled in Figure 1R-S; area units are misspelled in Figure 1E.

In this figure each graph has a different font styles and sizes.

2) Is Plum required for Myo signaling in the wing disc? Figure 2 should be expanded to directly probe for a role for Plum.

3) Provide higher magnification panels for Figure 4A-B. The current images and their low magnification are not enough to support the authors' arguments.

4) Please correct the following sentences in the subsection “Overexpression of Myo in muscle or fat body rescues *myo* mutant disc growth":

“First, ubiquitous overexpression of *myo* in wild type larva results in wings that were larger than normal (Figure 5—figure supplement 1). Next, we examined whether overexpression of *myo* in either glia, muscle, or fat body is able to rescue the size of *myo* mutant discs.”

5) Figure 5F is disproportionate and mislabeled.

6) Previous studies showed that the wing discs secretes Dpp in the hemolymph; Dpp reaches the ring gland and functions to control developmental timing.

Did the authors observe any disruption of the developmental progression in *myo* null mutants? If yes, they should speculate on the larger picture of inter-organ coordination.

Reviewer #2:

In this manuscript, Upadhyay et al. characterized the function of an Activin family member, Myoglianin (Myo), in *Drosophila* imaginal disc growth. They found that Myo uses the canonical *dSmad2*-mediated signaling to regulate wing size, and that a specific isoform of the Type-I receptor, Babo-A, is the sole signaling receptor in the wing disc mediating both canonical and non-canonical signaling. In addition, Punt and Plum function as co-effectors for the non-canonical Babo activity. The authors presented clear genetic and phenotypic data showing that Myo, which is mainly derived from muscle, can stimulate imaginal disc growth in a tissue non-autonomous manner.

This is an interesting study demonstrating the inter-organ signaling in growth control. The authors also carefully teased apart the signaling pathway mediating Myo function using genetic analysis. Experiments are well designed and most of the data support the conclusions. I have the following major comments:

1) The *myo* mutant imaginal discs are 40-50% smaller than controls. However, the *myo* mutant cells are only 22% smaller than the controls. The authors did not detect any difference in cell proliferation and apoptosis in fixed tissues, and suggested that there were slight differences overtime. It would be nice to perform live imaging of fluorescently labeled cells in imaginal discs incubated in vitro to support the authors' hypothesis.

2) It appears that Myo and its mouse homologs have opposite functions in regulating muscle size. What accounts for the difference? Would overexpressing Myo in mouse C2C12 cells promote muscle growth?

3) The authors suggested the activation of compensatory pathways in the null mutant, which may account for the differences in the null mutant vs. RNAi phenotypes. What is the muscle size phenotype when both *myo* and Actβare mutated?

4) Does knocking down *babo-a* lead to a decreased level of phosphorylated *dSmad2*?

5) Why knocking down *babo-a* by the Hh-GAL4 driver disrupts the A-P compartment boundary?

Reviewer #3:

How organ growth and sizes are coordinated within an organism during its development is a fascinating area of study that is not well understood. This emerging area clearly has important ramifications for our understanding of normal development and disease states, as well as for treatments achieved through regenerative medicine.

In this manuscript, the authors describe their work showing that the *Drosophila* Activin-like ligand, Myoglianin, regulates growth of the imaginal discs during development. Using a previously published myoglianin (*myo*) mutant, they show that the wing (and other) discs are between 40-50% smaller than control in this mutant. Loss of other Activin-like ligands, Dawdle and ActivinB, do not show the same affect. Myoglianin's effect on the discs appears to be through cell size regulation, as other parameters that could contribute to disc size (cell proliferation, cell death), do not appear to be affected in the *myo* mutant. The authors also report changes in the sizes of other cell types, specifically the larval somatic musculature, that is linked to loss of myoglianin. Note that this particular finding, smaller muscles, is a finding that is contrary to previous RNAi studies which indicated that loss of myoglianin lead to larger muscles.

The authors then investigate how Myoglianin mediates its effects. They show that Myoglianin signals in the wing disc through its receptors Baboon isoform A and Punt, and the accessory protein, Plum. These in turn lead to the phosphorylation of *dSmad2* and the subsequent regulation of gene transcription. Lastly, the authors investigate the source of Myoglianin. Expression of myoglianin is detected primarily in glia and muscle, although low levels are reported in the wing disc. Tissue specific knockdown using myoglianin RNAi suggest that muscle is the primary source of secreted myoglianin. Likewise rescue experiments in the *myo* mutant show that expression of myoglianin from the muscle yields complete rescue of the wing disc size; expression from glia can give some rescue and ectopic expression in fat body also lead to complete rescue.

This is an interesting study of inter-organ communication. It identifies a pathway that regulates organ size through regulation of cell size (myoglianin-Babo-a-Punt-Plum-dSmad). It challenges published studies on the impact of myoglianin on muscle size. Moreover the authors implicate the muscle as one source of signals that could regulate organ size.

Nevertheless, there remains several issues with the manuscript as submitted.

The major concerns:

1) For several of the points, muscle size-area and, given the changes in wing disc cell volume, volume-- should be assessed. For example, the authors should show that *myo* kd in muscle shows a larger muscle size as published. The authors should also test how changes in muscle size could influence *myo* levels and in turn, wing disc size.

2) Myoglianin level assessment; Could myoglianin levels in the hemolymph be measured under the different conditions?

3) Quantification: For all figures, the quantification should be improved. This is particularly true for Figure 4, all wing disc sizes should be quantified. In all cases N values, number of replicates should be reported in the figure legends. Better/higher mag images of *myoGAL4>UAS*-reporter should be included.

4) Consideration of the Adepithelial cells of the wing disc. The muscle progenitors that go on to populate the thoracic muscles are situated on the wing disc and could regulate its size. These express DMef2 and could also possibly be a local source for myoglianin. Specific GAL4 lines are available which drive expression in these. Having these as a local source of myoglianin would be an interesting finding.

There are additional issues with the figures listed below.

However, there are some common themes:

1) Control for RNAi usually is expression of an RNAi directed to GFP or mCherry. Controls for overexpression are usually expression of *UAS-GFP* or mCherry.

2) Better labeling of the images and include quantification please.

---

## [Author Response]

Reviewer #1:[…] The manuscript is compelling and clearly written. The findings are interesting and timely, especially as more and more reports find that inter-organ signaling control multiple aspects of development. However, I do have a number of concerns that need to be addressed before this manuscript could be considered for publication.First, the authors need to sort through some conflicting data published by other laboratories. However, they stop short of evaluating the reproducibility of some of these previous findings. Without turning this manuscript into a "set the record straight mission", I do think that this group of authors should use this stage to set the record straight and, in the process, validate some reagents or raise the appropriate flags. This process will benefit of the entire signaling community, especially since Myo has been the topic of extensive controversies.For example, the authors write a Discussion section on Myo as being functionally distinct from vertebrate Mstn. While I agree with the authors' interpretation, as a reader I really want to know whether the myo null and the muscle knockdown of myo indeed show reproducible opposing muscle phenotypes; I want to be aware of such extreme off-targets or compensatory effects. The authors have already validated a myo-RNAi line and used it, including in the muscle (Figure 4). They should complete the comparison and include/discuss their data.

We obtained the *myo* RNAi line used by (Augustin et al., 2017) and repeated their experiments. We found that knockdown of *myo* using *mef2-GAL4* does not significantly increase muscle size relative to *UAS-myoRNAi* alone. We controlled for gender in our analysis and only measured the muscle size of females. It is well appreciated in the field that females are substantially larger than the males and perhaps this could account for some differences that have been documented in prior studies. We also note that the genetic background of various GAL4 and UAS lines can significantly impact results. Specifically, we find that the *UAS-myoRNAi* line has significantly smaller muscles compared to any other “wild-type” controls that we have used. Therefore choosing such a line as a control “baseline” might lead to false interpretation of how *myo* affects muscle size.

Ideally, RNAi lines are validated by comparing their phenotypes to the genetic nulls. To validate whether *myo* regulates muscle size, we generated new alleles. Our previously published *myo* alleles (*myo1* and *myo4*) are on the same genetic background that can’t be “cleaned” because the 4th chromosome doesn’t recombine. For the new *myo* alleles we used CRISPR/Cas9 to generate targeted deletions or indel frameshifts of the *myo* coding sequence. Using a transheterozygous combination of the Crispr generated CR2 allele over *myo1* greatly reduces the possibility of 4th chromosome genetic background effects arising in the original alleles. We find that there is no difference in muscle size of *myo1/myoCR2* nulls versus heterozygous controls. It is important to reiterate that we have never observed an increase in muscle size using any combination of the genetic null alleles. Furthermore, knocking down *myo* ubiquitously using *da-GAL4* does not increase muscle size beyond that of *w1118* (Kim et al. BioRxiv) or *myo* hets or nulls (this report). Taken together our data suggests that *myo* does not regulate the size of larval body wall muscles. One last note is that we have completed a more thorough analysis of muscle size in all ligand mutants as well as *babo* and *dSmad2* nulls where we counted Z disc number in addition to measuring surface area. This helps control for potential differences in “stretching” during sample preparation. This analysis is available as a BiorXiv preprint (Kim and O’Connor, 2020, see Figure 1A, B, D).

Secondly, Myo was previously shown to bind to Babo and Wit in mammalian tissue culture. Nonetheless, the authors demonstrate that Wit is dispensable for Myo signaling in the wing disc. Based on the non-canonical signaling analysis, the authors argue that Myo signals through Babo-A, Punt and Plum. The authors should cement this conclusion using signaling assays which could presumably recapitulate and illustrate both canonical and non-canonical signaling pathways downstream of Myo. Such results will tremendously strengthen the authors' proposed model and will increase the overall impact of the paper. Since, so far, it's been difficult to reconstitute signaling by Drosophila activin-type ligands in tissue culture, the field will be urged to take note of the co-receptor contributions.

We would love to have a cell culture signaling assay that works for Myo and dAct. We have no trouble detecting signaling by the BMP ligands Gbb, Dpp, Scw and various heterodimers, but the only Activin-like ligand that signals reliably in S2 cells is Daw. We have manipulated the *babo-a* levels in S2 cells and checked for *plum* expression (which is strong) but we obtain no signal above background when Myo is added. We have tried numerous perturbations over the last 15 years and never gotten it to work. We suspect some other cofactor, or an unknown Myo activation step, is needed and there is no way to predict how long that might take to work out.

Thirdly, the data presented here do not exclude an alternative explanation: That the positive effect of Myo onto the wing growth is due to both canonical and non-canonical signaling through Babo-A. The authors should include pMad analyses in Figure 1T to address this possibility. Also, they should provide pMad and pSmad2 panels for Figure 5—figure supplement 1.

To address the reviewer’s concern, we knocked down *babo-a* and *plum* in the entire wing disc using a compound stock of *ci-GAL4* and *hh-GAL4*, then checked for p-dSmad2 and p-Mad levels in wing disc tissue extracts by western blotting. As a control, we crossed the GAL4 line to *w1118, UAS-iRFP, and UAS-iGFP*. Oddly, *Disc>iGFP* animals are sick and don’t make it past the L2 stage, and thus they were excluded from the analysis. (this illustrates that iGFP is not always a good control). Knocking down *babo-a* in the disc resulted in almost complete loss of p-dSmad2 levels. Knocking down *babo-a* did not alter p-Mad levels in the disc. Taken together, this demonstrates that Myo signaling via Babo-A activates *dSmad2* in the disc but not Mad. The data is now shown in Figure 2S-T. The suggestion to check p-Smad levels upon muscle overexpression of *myo* is interesting however we regret that we missed this request. For p-Mad, we don’t expect there to be a change. This is based on several literature reports that Mad phosphorylation by Babo in the wing disc only occurs under extreme (strong overexpression of activated *babo*) conditions, as well as the lack of P-Mad change in the *babo* lof discs described above. Given the current stay at home order, and that these are relatively minor points, we hope that this will be acceptable.

Other concerns:1) "muscle" is misspelled in Figure 1R-S; area units are misspelled in Figure 1E.In this figure each graph has a different font styles and sizes.

Corrected.

2) Is Plum required for Myo signaling in the wing disc? Figure 2 should be expanded to directly probe for a role for Plum.

We checked by western blot whether knockdown of *plum* resulted in a decrease in p-*dSmad2* levels in the wing disc. To our surprise, we did not detect any decrease. We are not able to reconcile this result with the original supplementary genetic data provided in the first submission that Plum was necessary for Myo signaling in the disc. It is possible that Plum is required with Babo-A to produce a non-canonical signal, but this requires more experimentation, which is not possible at the moment. Therefore, we removed the section about *plum* since it is not that important for the central message. We hope to design additional experiments in the future to more rigorously test the genetic interaction of *plum* with Myoglianin signaling in the wing disc and if we find something interesting, we could submit a short supplement describing these studies as a “Research Advances” to *eLife*.

3) Provide higher magnification panels for Figure 4A-B. The current images and their low magnification are not enough to support the authors' arguments.

The additional images are now provided as Figure 4—figure supplement 1.

4) Please correct the following sentences in the subsection “Overexpression of Myo in muscle or fat body rescues myo mutant disc growth”:“First, ubiquitous overexpression of myo in wild type larva results in wings that were larger than normal (Figure 5—figure supplement 1). Next, we examined whether overexpression of myo in either glia, muscle, or fat body is able to rescue the size of myo mutant discs.”

Sentence corrected.

5) Figure 5F is disproportionate and mislabeled.

Corrected.

6) Previous studies showed that the wing discs secretes Dpp in the hemolymph; Dpp reaches the ring gland and functions to control developmental timing.Did the authors observe any disruption of the developmental progression in myo null mutants? If yes, they should speculate on the larger picture of inter-organ coordination.

*Myo* mutants do have developmental progression defects. They appear to stall at late 3rd instar larval stage and fail to undergo proper pupariation. They do not wander and form prepupa that are often curved or elongated on the food surface. These prepupa do not undergo head eversion and fail to differentiate disc derived structures. This phenotype is due to a defect in glia to neuron signaling since it can be rescued by Myo expression in the surface glia (unpublished). We are currently exploring this phenotype in more detail for a future report.

Reviewer #2:[…] This is an interesting study demonstrating the inter-organ signaling in growth control. The authors also carefully teased apart the signaling pathway mediating Myo function using genetic analysis. Experiments are well designed and most of the data support the conclusions. I have the following major comments:1) The myo mutant imaginal discs are 40-50% smaller than controls. However, the myo mutant cells are only 22% smaller than the controls. The authors did not detect any difference in cell proliferation and apoptosis in fixed tissues, and suggested that there were slight differences overtime. It would be nice to perform live imaging of fluorescently labeled cells in imaginal discs incubated in vitro to support the authors' hypothesis.

In theory this is a good idea. However, to our knowledge, only Suzanne Eaton’s group has been able to grow mid third instar discs in culture for up to 24 hr and this time frame may still not be long enough to reliably detect a subtle growth rate change (Dye et al. 2017 Dev.) To set up a live imaging system in our lab and to implement the segmentation program necessary to follow individual divisions on our scope would take many months and it still may be difficult to detect the changes. Therefore, we can’t do this experiment at this time. However, we do note that Hevia et al., 2017, did measure *babo* and *dSmad2* null mutant clone sizes in the wing discs and found them to be smaller than the wildtype twin spots. Their interpretation was that loss of *dSmad2* caused a 2 fold proliferation defect. Since, as we show in our report, that Myo is responsible for all of the p-dSmad2 signal in the wing disc, we think this data, coupled with the cell size reduction that we report, is sufficient to explain the smaller wing disc size of *myo* mutants.

2) It appears that Myo and its mouse homologs have opposite functions in regulating muscle size. What accounts for the difference? Would overexpressing Myo in mouse C2C12 cells promote muscle growth?

We are not sure why there is a difference in the direction of the effect on muscle size between the vertebrate and the fly. We have done an extensive analysis of how Activin influences muscle size and have a drafted a paper (Kim and O’Connor, 2020 BioRiv) which demonstrates that Activin regulates muscle size, in part, through its effects on Akt1 and Pdk1 expression. Mammalian myostatin is also thought to regulate muscle size, in part, by its effects on the IsR/Tor pathway, but it appears to be negative instead of positive. We discuss this issue extensively in the Kim and O’Connor manuscript and briefly refer to it in the Discussion of the present manuscript. As to adding Myo ligand to PC12 cells, it will likely not work since there is no Babo-A in these cells and the vertebrate Smads 2/3 are very different in sequence within the Mh1 DNA binding domain compared to *Drosophila* dSmad2. Eventually, it may be worth trying to recapitulate the entire *Drosophila* pathway, including expression of *dSmad2*, in a vertebrate cell line to see if and how it affects Ins/Tor signaling, but until we have a better understanding of the important variables in a *Drosophila* cell culture system, this type of experiment will likely not uncover any useful insights into why the effects on Ins/Tor pathway are opposite in the two systems.

I would also like to point out that the present paper is not about muscle size control. It is about inter-organ signaling and Myo control of disc growth. The muscle aspects are really a side issue, and as mentioned above, are the subject of a much more extensive analysis that has been submitted as a separate manuscript and is available at BioXriv.

3) The authors suggested the activation of compensatory pathways in the null mutant, which may account for the differences in the null mutant vs. RNAi phenotypes. What is the muscle size phenotype when both myo and Actβare mutated?

Again, not the primary subject of this paper, but yes, we have made the double mutant and the muscle is still small. A more extensive analysis of the phenotypes produced by all double mutant combinations as well as he triple mutant is now underway but in no case do we see larger muscles.

4) Does knocking down babo-a lead to a decreased level of phosphorylated dSmad2?

Knocking does lead to decreased levels of p-dSmad2 in the wing disc. This result is now shown in Figure 2S.

5) Why knocking down babo-a by the Hh-GAL4 driver disrupts the A-P compartment boundary?

We do not know, but we are exploring whether other signaling pathways are affected when Activin signaling in the disc is disrupted.

Reviewer #3:[…] This is an interesting study of inter-organ communication. It identifies a pathway that regulates organ size through regulation of cell size (myoglianin-Babo-a-Punt-Plum-dSmad). It challenges published studies on the impact of myoglianin on muscle size. Moreover the authors implicate the muscle as one source of signals that could regulate organ size.Nevertheless, there remains several issues with the manuscript as submitted.1) For several of the points, muscle size-area and, given the changes in wing disc cell volume, volume-- should be assessed. For example, the authors should show that myo kd in muscle shows a larger muscle size as published. The authors should also test how changes in muscle size could influence myo levels and in turn, wing disc size.

This issue of why our results differ from published RNAi studies was addressed above. We do not see the larger larval muscle size that was reported by Augustine et al., 2017 when *myo* is specifically knocked down in muscles. We have looked a bit at how muscle size affects wing discs, but it is very complicated. It is not likely that it is size per se, but some other aspect of muscle physiology. Our preliminary data suggest that the results are strongly dependent on the nutrient composition of the media that the larvae feed on. Again, this will require a detailed analysis, including a study of how the *myo* promoter/enhancer is being regulated by various environmental manipulations. This is in progress, but is beyond the scope of the present study.

2) Myoglianin level assessment; Could myoglianin levels in the hemolymph be measured under the different conditions?

We have not tried, but other mass spec results only report finding Daw in hemolymph. So far, our attempts to tag Myo and produce an active ligand have been unsuccessful. As a result, we have not further pursued analysis of Myo in hemolymph.

3) Quantification: For all figures, the quantification should be improved. This is particularly true for Figure 4, all wing disc sizes should be quantified. In all cases N values, number of replicates should be reported in the figure legends. Better/higher mag images of myoGAL4>UAS-reporter should be included.

Quantification for Figure 4 is now shown in Figure 4L. The higher magnification of Figure 4A and split channel images for Figure 4B are shown in Figure 4—Figure supplement 1.

4) Consideration of the Adepithelial cells of the wing disc. The muscle progenitors that go on to populate the thoracic muscles are situated on the wing disc and could regulate its size. These express DMef2 and could also possibly be a local source for myoglianin. Specific GAL4 lines are available which drive expression in these. Having these as a local source of myoglianin would be an interesting finding.

This is a good point. We initially did consider that the adepithelial myoblasts in the wing disc could provide the Myo necessary for disc growth. However, two sets of data suggest that this is not the case. First, as shown in Figure 4, we find that knocking down *myo* in the muscle using *mhc-GAL4* is sufficient to rescue the *dSmad2* wide wing phenotype. Unlike *mef2-GAL4,* the *mhc-GAL4* line is not expressed in the adepithelial myoblasts of the wing disc. Therefore, in animals where *myo* is knocked down by *mhc-GAL4*, Myo should still be secreted by the myoblasts, and yet we observe a complete rescue of the *dSmad2* wide wing phenotype. We also examined adepithelial cells for *myo* expression using our *myo-GAL4* reporter line and by in situ hybridization. In neither casedid we observe any expression. Taken together, we believe these observations indicate that the adepithelial cells are not the source of Myo that regulates disc growth.

There are additional issues with the figures listed below.However, there are some common themes:1) Control for RNAi usually is expression of an RNAi directed to GFP or mCherry. Controls for overexpression are usually expression of UAS-GFP or mCherry.

When doing new experiments for the western blots in Figure 2 we did try the *UAS-iGFP* and *UAS-iRFP* lines. As mentioned the iGFP line was toxic and we were never able to dissect wing discs from 3rd instar larvae. One could argue that the best control for RNAi experiments is a scrambled sequencewhich is just not feasible for every experiment.

2) Better labeling of the images and include quantification please.

Corrected.